# Distinct recruitment of dorsomedial and dorsolateral striatum erodes with extended training

**Youna Vandaele[1]\*, Nagaraj R Mahajan[2], David J Ottenheimer[3], Jocelyn M Richard[4], Shreesh P Mysore[1,3], Patricia H Janak[1,3,5]\***

[1]Department of Psychological and Brain Sciences, Krieger School of Arts and Sciences, Johns Hopkins University, Baltimore, United States; [2]Department of Electrical and Computer Engineering, Johns Hopkins University, Baltimore, United States; [3]The Solomon H. Snyder Department of Neuroscience, Johns Hopkins School of Medicine, Johns Hopkins University, Baltimore, United States; [4]Department of Neuroscience, University of Minnesota, Minneapolis, United States; [5]Kavli Neuroscience Discovery Institute, Johns Hopkins University, Baltimore, United States

**\*For correspondence:**
youna.vandaele@jhu.edu (YV);
patricia.janak@jhu.edu (PHJ)

**Competing interests:** The authors declare that no competing interests exist.

**Abstract** Hypotheses of striatal orchestration of behavior ascribe distinct functions to striatal subregions, with the dorsolateral striatum (DLS) especially implicated in habitual and skilled performance. Thus neural activity patterns recorded from the DLS, but not the dorsomedial striatum (DMS), should be correlated with habitual and automatized performance. Here, we recorded DMS and DLS neural activity in rats during training in a task promoting habitual lever pressing. Despite improving performance across sessions, clear changes in corresponding neural activity patterns were not evident in DMS or DLS during early training. Although DMS and DLS activity patterns were distinct during early training, their activity was similar following extended training. Finally, performance after extended training was not associated with DMS disengagement, as would be predicted from prior work. These results suggest that behavioral sequences may continue to engage both striatal regions long after initial acquisition, when skilled performance is consolidated.

DOI: https://doi.org/10.7554/eLife.49536.001

## Introduction

The dorsal striatum plays a pivotal role in learning and performing actions, including sequential actions, made to obtain rewarding outcomes, with distinct functions proposed for different striatal subregions (*Balleine et al., 2009*; *Graybiel and Grafton, 2015*). The dorsomedial striatum (DMS) receives excitatory inputs from prefrontal cortices and is implicated in goal-directed action control, whereas the dorsolateral striatum (DLS) primarily receives inputs from sensorimotor and premotor cortices and is implicated in habit and skill learning (*Balleine et al., 2009*; *Corbit and Janak, 2016*; *Corbit et al., 2007*; *Corbit and Janak, 2010*; *Graybiel and Grafton, 2015*; *Yin and Knowlton, 2006*; *Yin et al., 2004*; *Yin et al., 2005*; *Yin and Knowlton, 2006*; *Yin et al., 2006*).

Previous work has examined neural activity within the DMS and DLS to determine how these regions might contribute to instrumental behavior and skill learning. When subjects are trained to execute a series of lever presses for reward, neuronal excitations emerge over learning in the dorsal striatum at the initiation and termination of the response sequence (*Jin and Costa, 2010*; *Jin et al., 2014*). These neural excitations have been termed start/stop responses, and, along with sustained

inhibitions and excitations, are proposed to reflect chunking of individual actions into behavioral units (*Jin and Costa, 2015*; *Jin and Costa, 2010*; *Jin et al., 2014*; *Martiros et al., 2018*). These sequence-related neural responses are reported to be more numerous in the DLS (*Martiros et al., 2018*). Consistent with this, contrasting patterns of neural ensemble activity emerge in the DMS and DLS as rats learn to choose the correct arm of a T-maze in response to sensory cue presentations (*Barnes et al., 2005*; *Thorn et al., 2010*), with task-bracketing activity emerging in the DLS (*Barnes et al., 2005*; *Jog et al., 1999*; *Smith and Graybiel, 2016*; *Thorn et al., 2010*), but not in the DMS. The DLS task-bracketing activity during navigational sequences is similar to that observed during acquisition of lever-pressing sequences and likewise may reflect chunking (*Redish, 2016*; *Smith and Graybiel, 2014*; *Smith and Graybiel, 2016*). However, the presence of task-bracketing neuronal activity around actions made to obtain reward is not always detected (*Sales-Carbonell et al., 2018*). More broadly, it is not clear how behavior-related neural activity in DMS and DLS relates to behavioral improvement over time.

Here we sought to characterize possible sequence-related neural correlates in the striatum within a reward-seeking task specifically chosen for its ability to promote rapid expression of automated and habitual lever pressing behavior (*Vandaele et al., 2017*). In this task, rats must wait for lever insertion after which they complete a series of five lever presses to obtain access to reward. Lever retraction occurs after the fifth lever press and signals reward delivery. Lever insertion and retraction thus constitute audio-visual stimuli signaling the opportunity for reward and its delivery, respectively. We previously showed that responding for sucrose reward rapidly becomes habitual when the lever insertion and retraction cues are provided in this discrete-trials fixed-ratio 5 (DT5) task; in contrast, responding remains sensitive to reward devaluation, a test of goal-directed control, in absence of these cues under a free-running fixed-ratio five schedule (*Vandaele et al., 2017*). Using the DT5 task, we sought here (1) to compare cue and action encoding in DMS and DLS during learning of the task and after extended training, (2) to characterize the dynamics of behavioral sequence-related neural activity in DMS and DLS within and across training stages, and (3) to examine the link between activity patterns in DLS and DMS with performance. As subjects showed increasing indicators of skilled performance, we expected to see the development of greater behavior-related activity in DLS versus DMS during training, and stronger correlation of DLS activity with indices of improved performance (*Jog et al., 1999*; *Regier et al., 2015*; *Thorn et al., 2010*). We also purposefully included a group of rats with very extensive training (>2 months) in which we expected to find that DLS sequence-related activity had strengthened over time, perhaps as consolidation of skills progressed (*Barnes et al., 2005*; *Smith and Graybiel, 2013*), leading to further dissociation of DLS and DMS activity.

Contrary to our expectations, we found sequence-related activity in both DLS and DMS, during both early and extended training. During early training, the nature of sequence-related activity was markedly different between DLS and DMS; DLS neurons were predominantly excited during the behavioral sequence whereas DMS neurons were predominantly inhibited, with excitation at the sequence boundaries. Further, many sequence-related firing patterns appeared to reflect stimulus attributes rather than motor initiation, in contrast to some prior reports (*Jin and Costa, 2010*; *Jin et al., 2014*) and in agreement with other work (*Sales-Carbonell et al., 2018*). Additionally, no substantial evolution in neural activity patterns was observed across early training sessions despite habitual learning and significant improvement in performance. Interestingly, however, the patterns of activity in the DMS and DLS were similar to each other after extended training, with a balanced distribution of task-related inhibition and excitation in both regions. Finally, the spike activity of a substantial proportion of neurons in both regions was correlated with specific aspects of performance on a trial-by-trial basis, indicating that optimized performance after extended training was not associated with DMS disengagement from behavioral control. Together, these findings suggest that both regions of the striatum are differentially, yet complementarily, involved during early training, but act in concert when series of actions are performed with great regularity after extended training.

## Results

### Graded performance improvement during early training and optimization of behavior after extended training in the discrete trials Fixed Ratio-5 (DT5) procedure

To characterize neural activity in DMS and DLS during early and extended training, rats were trained in a discrete trials fixed ratio-5 (DT5) procedure. We have previously shown that rats trained in this procedure rapidly develop automated lever pressing behavior that is relatively insensitive to reward devaluation, a finding taken to indicate habitual control (*Vandaele et al., 2017*). In this task, each trial began with lever insertion after which rats were required to complete a sequence of 5 lever presses to obtain access to reward, signaled by lever retraction after the fifth lever press (*Figure 1A*). Rats were given the opportunity to respond on 30 trials separated by 1 min intertrial intervals. To measure neural activity during learning, one group of rats was implanted with fixed recording electrodes in the DMS and DLS before training in the DT5 task, and neural activity in these regions was recorded during acquisition (early training group, N = 9). To measure neural activity in well-trained subjects, rats in the second group were implanted in the DMS and DLS with drivable recording electrodes after more than 8 weeks of training in the DT5 procedure (extended training group, N = 8; *Figure 1B*). Throughout training, lever pressing remained near the maximum possible (150 responses/session) (*Figure 1C*), yet performance become more automated across sessions in the early training group, indicative of sequence learning (*Figure 1D–G*). Specifically, the number of within-sequence reward port entries was rapidly suppressed across sessions (Friedman ANOVA $\chi^2$ = 44.82, p<0.0001; *Figure 1D*), including in the first DT5 session, as subjects learned the response requirement had increased from one to five. Learning was also accompanied by a progressive increase in response rate (*Figure 1E*; $F_{(9,72)}$=16.59, p<0.0001) and a concomitant decrease in trial-by-trial variability in within-sequence response rate (*Figure 1F*; $F_{(9,72)}$=3.12, p<0.01). The latency to first lever press also decreased across early training sessions (*Figure 1G*; $F_{(12,96)}$=10.87, p<0.0001). Following extended training in the second group of rats, within-sequence port entries did not differ from zero and response rate, trial-by-trial variability in response rate and the latency to first lever press reached a plateau (*Figure 1C–G*). For both early and extended training groups, we assessed sensitivity to outcome devaluation at the end of recording, using sensory-specific satiety (*Figure 1H–I*). While mean responding decreased following devaluation, this decrease was not significant after either early or extended training when analyzing the number of lever presses (*Figure 1H*; early training: $F_{(1,8)}$=4.87, p=0.06; extended training: $F_{(1,7)}$=3.75, p=0.09) or the number of trial initiated (*Figure 1I*; early training: $F_{(1,8)}$=4.25, p=0.07; extended training: $F_{(1,7)}$=3.5, p>0.1), suggesting that behavior was under habitual control, although there was variability within the groups.

### DMS and DLS neurons are differentially modulated in the DT5 procedure during early training but not after extended training

During acquisition of the task in the early training group, we recorded an average of 37 (±1.3) and 81 (±1.8) units per session in the DLS and DMS, respectively, across the entire group of rats. The number of recorded units remained stable across sessions (*Figure 2—figure supplement 1A*). In the extended training group, by advancing the electrodes every other day, a total of 387 and 462 neurons were recorded in DLS and DMS, respectively (*Figure 2—figure supplement 1B*). Electrode placements were similar in the early and extended training groups (*Figure 2—figure supplement 1C*). To focus on the primary population of striatal neurons, putative medium spiny neuron (MSN) and interneuron populations were distinguished using firing rates and waveform properties (*Figure 2—figure supplement 1D–F*; *Martiros et al., 2018*; *Schmitzer-Torbert and Redish, 2008*; *Stalnaker et al., 2016*). Putative MSNs represented 95% and 88% of the recorded units in the early and extended training groups, respectively. Fast-spiking interneurons (FSI) were extracted based on their firing rate and half-width valley (*Figure 2—figure supplement 1D–F*), and represented on average 1.1% of units in the early training group (0–3 neurons per session) and 4.1% of units in the extended training group (N = 31). Neurons not classified as putative-FSI but showing features intermediate to MSN and FSI were unclassified (early training, N = 1–9 per session; extended training, N = 74). As reported previously, we could not reliably isolate putative tonically active neurons

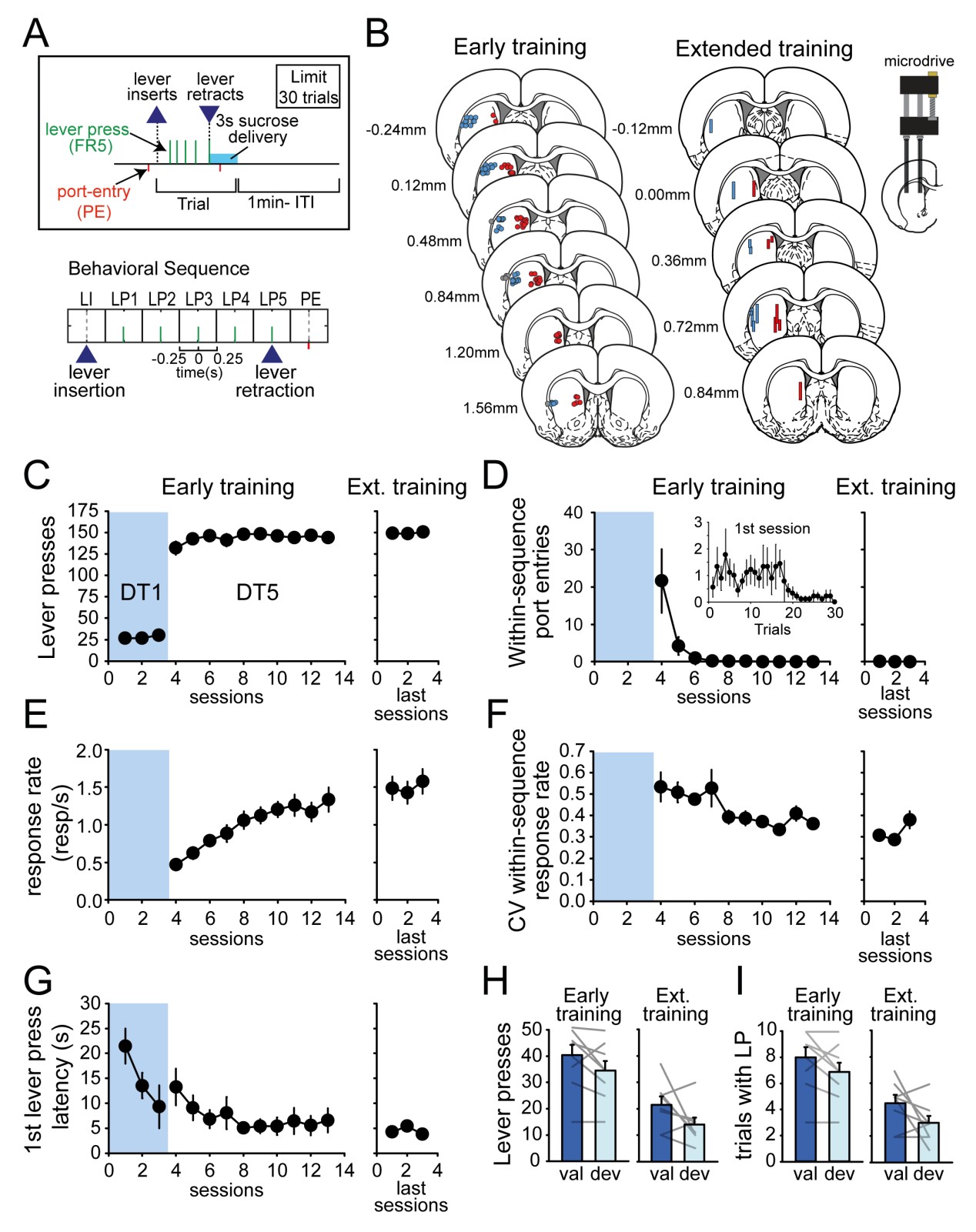

**Figure 1.** Rats rapidly develop habit and automaticity in the DT5 task. (A) Schematic of the DT5 task and event-related time intervals considered for analyses. (B) Histological reconstruction of electrode placements in DLS and DMS. (C–G) Number of lever presses (C), within-sequence port entries (D), response rate (responses per second) (E), coefficient of variation of within-sequence response rate (F), and first lever press latency (G) across early training and recording sessions (3 DT1 sessions followed by 10 DT5 sessions) and during the last three recording sessions in the extended training

*Figure 1 continued on next page*

*Figure 1 continued*

group. Inset in D: Number of within-sequence port entries across trials during the first DT5 session. (H–I) Satiety-induced devaluation: number of lever presses (H) and number of initiated trials (I) in the valued (val) and devalued (deval) conditions under extinction after early training (left) and extended training (right). Error bars show SEM.

DOI: https://doi.org/10.7554/eLife.49536.002

(TANs) based on waveform (*Sales-Carbonell et al., 2018*). Given the low number of TANs in dorsal striatum (about 1%; *Oorschot, 2013*; *Schmitzer-Torbert and Redish, 2008*), we assumed any possible misclassification of TANs as MSNs, though unlikely, would have minimal impact on the population characteristics presented in this study. In this paper we focus only on the putative MSN population.

On average, 78.2% (±1.4) of putative MSNs from the early training dataset showed a significant change in firing rate to one or more events, and were termed task-responsive neurons (TRN) (*Figure 2A*). Among them, a majority of DLS neurons increased their spike activity during the behavioral sequence (excitation, range N = 15–24 per session; inhibition, range N = 4–9 per session) whereas DMS neurons predominantly decreased their spike activity (excitation, range N = 13–21 per session; inhibition, range N = 34–61 per session). The relative proportions of excitations and inhibitions significantly differed between DLS and DMS for the ten sessions analyzed during early training (*Figure 2A*; DLS vs DMS, $\chi^2$-values > 7.2, p-values<0.01).

In the extended training dataset, 76.5% of the analyzed neurons were task-responsive. In clear contrast with the early training group, the number of neurons excited and inhibited during the behavioral sequence was similar in DLS versus DMS (*Figure 2A*; DLS excitation, N = 143; DLS inhibition, N = 109; DMS excitation, N = 158; DMS inhibition, N = 162; DMS vs DLS: $\chi^2$ = 3.07, p>0.05).

To investigate in more detail the presence or absence of regional differences in behavior-related spiking activity of putative-MSNs, we examined the normalized firing across the trial, focusing on seven consecutive events of the behavioral sequence (*Figure 1A*; lever insertion, each of 5 lever presses, and first port-entry following reward delivery) in the early and extended training groups, combining all TRNs (*Figure 2B-E*). During early training, we observed strong differences between the activity of TRN in DLS and DMS along the behavioral sequence. On average, the DLS population showed a sustained increase in activity throughout the behavioral sequence whereas the mean activity for the DMS population was increased at the sequence boundaries and decreased during lever presses (*Figure 2A–B*). Repeated measures ANOVA revealed differences between regions and across events (*Figure 2B and D*; main effect of region, $F_{(1,839)}$=135.1, p<0.0001; region * event interaction, $F_{(11,9229)}$ = 30.85, p<0.0001). When mean neural activity was examined after extended training, the differences in DMS and DLS population activity were not significant, with no effect of region, nor a region by event interaction (*Figure 2C and E*; region $F_{(1,570)}$=2.53, p>0.1, region*event $F_{(11,6270)}$ = 1.48, p>0.1). Thus the mean activity of TRNs differed between DMS and DLS during early training, but not after extended training.

Surprisingly, there was no significant change in mean normalized activity in DMS and DLS across early training sessions despite significant performance improvements during the task (*Figure 1D–G*; *Figure 2B and D*, top versus bottom panels; main effect of session, $F_{(9,839)}$=0.49, p>0.1; session*event interaction, $F_{(99,9229)}$ = 0.72, p>0.1). We therefore looked earlier in training to the period of transition between the DT1 sessions (one press required) and the DT5 sessions (five presses required). Here we observed an increase in DLS neural activity in the 500 ms before the first lever press across DT1 sessions (*Figure 2—figure supplement 2*; Wilcoxon test, 1st vs 3rd DT1: Z = −2.29, p<0.05; 1st DT1 vs 1st DT5: Z = −2.16, p<0.05). Within the first DT1 session, DLS activity on average increased at lever insertion and then returned to baseline before the first lever press; the activity was sustained between lever insertion and first lever press during subsequent sessions (*Figure 2—figure supplement 2A–C*). This early change may reflect shortening of the first lever press latency occurring across the first three DT1 sessions (*Figure 2—figure supplement 2D–G*, *Figure 1G*), as rats learned about the task. However, we still expected neuronal changes correlated with the improvement in performance, once the response requirement extended to five lever presses. Yet, the average activity in dorsal striatum did not change in step with behavior as subjects' performance improved during

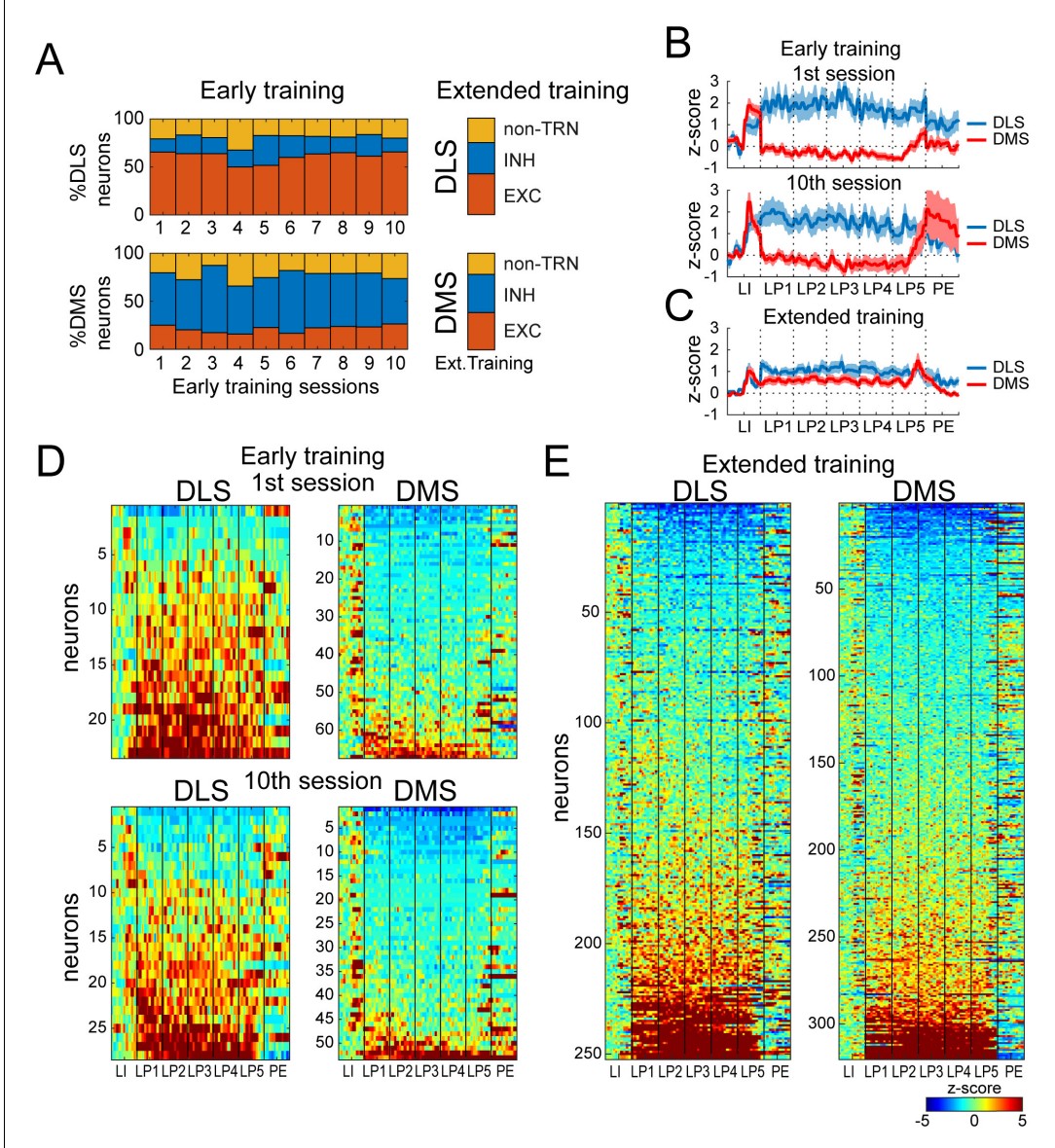

**Figure 2.** DMS and DLS neurons are differentially modulated in the DT5 task during early training but not extended training. (A) Proportion of non-task-responsive neurons (non-TRN), and task-responsive neurons (TRN) excited (EXC) or inhibited (INH) during the sequence, across early training sessions (left) or after extended training (right) in DLS (up) and DMS (bottom). (B–C) Mean z-score (± SEM) of DLS (blue) and DMS (red) TRNs during (B) early training sessions day 1 (top) and 10 (bottom) (N = 9 animals) and (C) during extended training (N = 8 animals). (D–E) Heatmaps of DLS (left) and DMS (right) TRNs, sorted by mean sequence-related activity, during (D) early training session 1 (top) and 10 (bottom) and (E) extended training.
DOI: https://doi.org/10.7554/eLife.49536.003

The following figure supplements are available for figure 2:

**Figure supplement 1.** Recording results summary and isolation of medium spiny neurons.
DOI: https://doi.org/10.7554/eLife.49536.004
**Figure supplement 2.** Moderate changes in DLS activity across the first DT1 early training sessions.
DOI: https://doi.org/10.7554/eLife.49536.005

the first 10 sessions of DT5 training. Furthermore, when comparing with recordings made after extended training, we observed more similarity between regions, rather than increasing difference in DLS and DMS activity over time with the optimization of behavior across repeated practice.

We next considered the possibility that averaging the activity of DMS and DLS neurons may have masked subtle regional differences in the activity patterns of individual neurons that may track

behavioral improvement. Indeed, as previously reported (*Jin et al., 2014*), we observed substantial variability in individual patterns of neuronal activity along the behavioral sequence (*Figure 2D–E*, *Figure 2—figure supplement 1G*). Investigating the distribution of distinct neural signatures among individual neurons in DLS versus DMS during early and extended training could, therefore, provide a more precise characterization of neural activity from which to examine possible differences in DMS and DLS activity across training stages.

## Classification of distinct neural signatures in the dorsal striatum during extended sequence training

To identify distinct neural signatures associated with performance of the DT5 lever pressing task we examined activity during a time period when behavior was very stable, after extended training. We applied an objective statistical approach of hierarchical clustering and a model selection metric to the extended training dataset containing putative MSNs from both DMS and DLS. First, we observed that many neurons showed transient changes in activity (<0.25 s; termed 'Phasic') while many others expressed excitations or inhibitions that were sustained in time (>0.5 s; 'Non-phasic'). We considered that these phasic and sustained activity neurons might have distinct relations to performance of the behavioral sequence; we were especially interested to examine neurons that appeared to have sustained activity given prior reports on these profiles and their proposed relation to action sequences (*Jin and Costa, 2015*; *Jin and Costa, 2010*; *Jin et al., 2014*). Therefore we sought to separate Phasic and Non-phasic neurons, using a Fourier analysis on the extended training dataset. We reasoned that sustained (Non-phasic) modulation of activity during the behavioral sequence should be associated with higher power in the low frequency region (<1 Hz), whereas transient (Phasic) modulations of sequence-related activity should result in higher power in the intermediate frequency region (1–4 Hz) (Materials and methods; *Figure 3—figure supplement 1*). Therefore, power in low (<1 Hz) and intermediate (1–4 Hz) frequency domains for each neuron were used as features for hierarchical clustering. Following hierarchical clustering on the combined DMS and DLS dataset, the optimal number of classes was determined using a model selection metric (the Calinski Harabasz criterion; CH index). This method resulted in two significant classes of neurons (*Figure 3—figure supplement 1*; permutation test, p<0.0001). The majority of neurons were characterized by transient peaks of activity at some time in the behavioral sequence, and were classified as Phasic neurons (N = 635; *Figure 3A*, *Figure 3—figure supplement 1*). Neurons from the other class mostly expressed sustained activity throughout the lever press sequence and were classified as Non-phasic neurons (N = 113; *Figure 3—figure supplement 1*). The proportion of Phasic versus Non-phasic neurons did not significantly differ between brain regions (*Figure 3—figure supplement 1C*; $\chi^2$ = 0.39, p>0.1).

Among Phasic neurons, we observed transient excitation at the start, in the middle, or at the end of the sequence (*Figure 3A*), and sought to examine if there were systematic regional differences in this activity. Hierarchical clustering applied to the combined set of DMS and DLS Phasic neurons resulted in the separation of 3 significant classes of neurons (*Figure 3B*): neurons expressing transient excitations after the lever insertion ('Start' neurons, DLS N = 46, DMS N = 64; *Figure 3C*), before the port entry ('Stop' neurons, DLS N = 72, DMS N = 91; *Figure 3D*) or modulations at one or more points during the lever presses ('Middle' neurons, DLS N = 172, DMS N = 190; *Figure 3E*). The proportions of neurons in each Phasic class and their average z-scores did not significantly differ between DLS and DMS ($\chi^2$ = 1.3, p > 0.1; region: F-values < 3.5, p-values > 0.05; region*events: F-values < 2.1, p-values > 0.05).

The classification analysis reveals that many Phasic neurons show activity near the start or end of the behavioral sequence, in agreement with prior descriptions of 'Start-Stop' related activity. Yet, it is unclear whether the 'Start' activity identified here (*Figure 3C*) represents a response to the reward-predictive cue of lever insertion or a 'Start' signal for the initiation of the sequence. To address this question, we examined the activity of Phasic Start neurons aligned to both lever insertion and the time of first lever press, and found on average that this population showed a peak in activity within 250msec after lever insertion, without a concomitant peak within the 250msec prior to the first lever press (*Figure 4A*). Statistical analysis of spiking activity revealed that only 15% of Phasic Start neurons (16/110 units) increase their spiking activity prior to the first lever press relative to baseline (i.e., at the presumed time of sequence initiation) (*Figure 4B*). These results suggest that, in

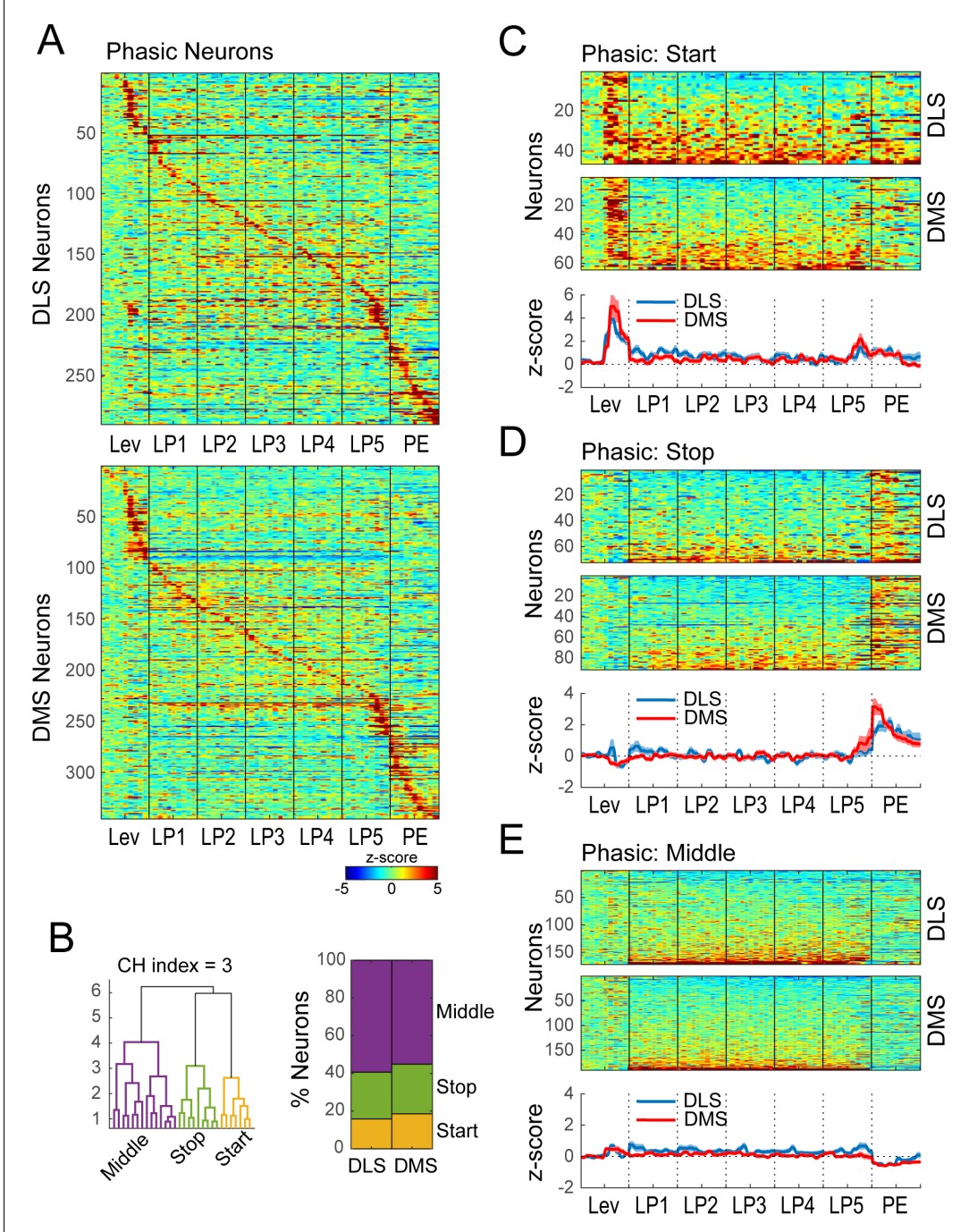

**Figure 3.** Hierarchical clustering on Phasic neurons during extended training. (A) Heatmap of normalized activity of neurons identified as being Phasic (290/338 neurons in DLS and 345/410 in DMS) sorted by time of peak activity. (B) Hierarchical clustering on Phasic neurons (combined DMS and DLS): dendrogram (left) and proportion of DLS and DMS neurons in the Start, Stop and Middle classes (right). (C–E) Heatmaps (top) and average z-score (± SEM) (bottom) of Phasic Start neurons (C; 46/290 and 64/345), Phasic Stop neurons (D; 72/290 and 91/345) and Phasic Middle neurons (E; 172/290 and 190/345) separated by region; neurons are sorted by averaged normalized activity during the behavioral sequence.

DOI: https://doi.org/10.7554/eLife.49536.006

The following figure supplement is available for figure 3:

**Figure supplement 1.** Fourier analysis and hierarchical clustering allowing separation of Phasic and Non-phasic neurons during extended training.
DOI: https://doi.org/10.7554/eLife.49536.007

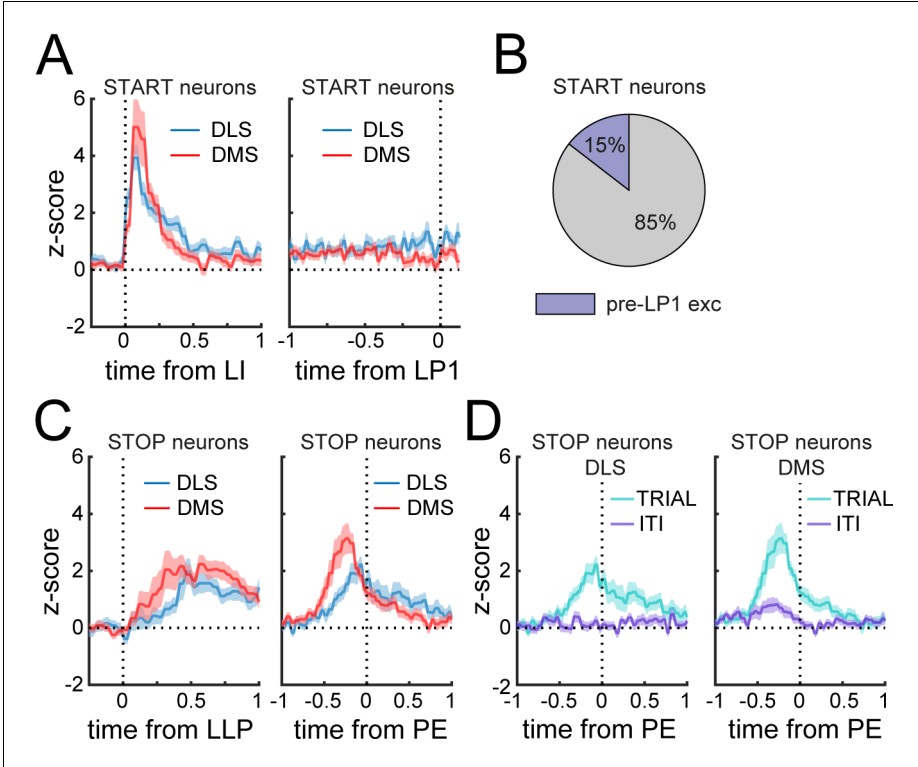

**Figure 4.** Phasic Start neurons do not encode the initiation of the sequence. (**A**) Mean z-score (± SEM) around lever insertion (left, LI) and first lever press (right, LP1) in DMS and DLS Phasic Start neurons. (**B**) Proportion of Phasic Start neurons increasing spiking activity before the first lever press (pre-LP1 exc) compared to baseline. (**C**) Mean z-score (± SEM) aligned to the last lever press (left, LLP) and port entry (right, PE) in DMS and DLS Phasic Stop neurons. (**D**) Mean z-score (± SEM) aligned to port entry in DLS (left) and DMS (right) Phasic Stop neurons at the termination of rewarded trials (TRIAL) or during inter-trial intervals (ITI).

DOI: https://doi.org/10.7554/eLife.49536.009

the main, the Phasic Start neurons responding to the lever insertion cue are not explicitly signaling motor sequence initiation.

Similarly, it is not clear whether Phasic Stop neurons signal the termination of the motor sequence, a response to the lever retraction sensory stimulus, expectation of reward, or a combination of these. Phasic Stop neurons increased activity after the last lever press with an even greater peak before the port entry (*Figure 4C*), that is, before consumption of the reward. This port entry peak was not observed during port entries occurring during inter-trial intervals (*Figure 4D*), suggesting that this activity may encode termination of the sequence and/or reward expectation during the port approach.

Among Non-phasic neurons, we observed either sustained excitation or sustained inhibition during lever presses (*Figure 5A*). To quantify this, Non-phasic neurons were further subdivided using hierarchical clustering. This analysis resulted in the separation of 2 significant classes of neurons (*Figure 5B–D*): neurons expressing sustained inhibition ('INH' neurons, DLS N = 17/48, 35%; DMS N = 36/65, 55%; *Figure 5C*) or sustained excitation during the five lever presses ('EXC' neurons, DLS N = 31/48, 64%; DMS N = 29/65, 45%; *Figure 5D*). The proportion of neurons across the Non-phasic classes did significantly differ between DLS and DMS ($\chi^2$ = 4.4, p<0.05) with a higher proportion of EXC neurons in DLS compared to DMS (*Figure 5B*). In each class, the average normalized activity of DMS and DLS neurons did not differ across events in the behavioral sequence (region*event, INH $F_{(11,561)}$ = 0.41, p>0.1, EXC $F_{(11,638)}$ = 1.56, p>0.1).

Overall, five similar firing patterns were identified after extended training that were each present in both DMS and DLS with a modest overrepresentation of sustained excitations (EXC) in DLS. Although these 5 classes of neurons were separated based on the most salient features of each

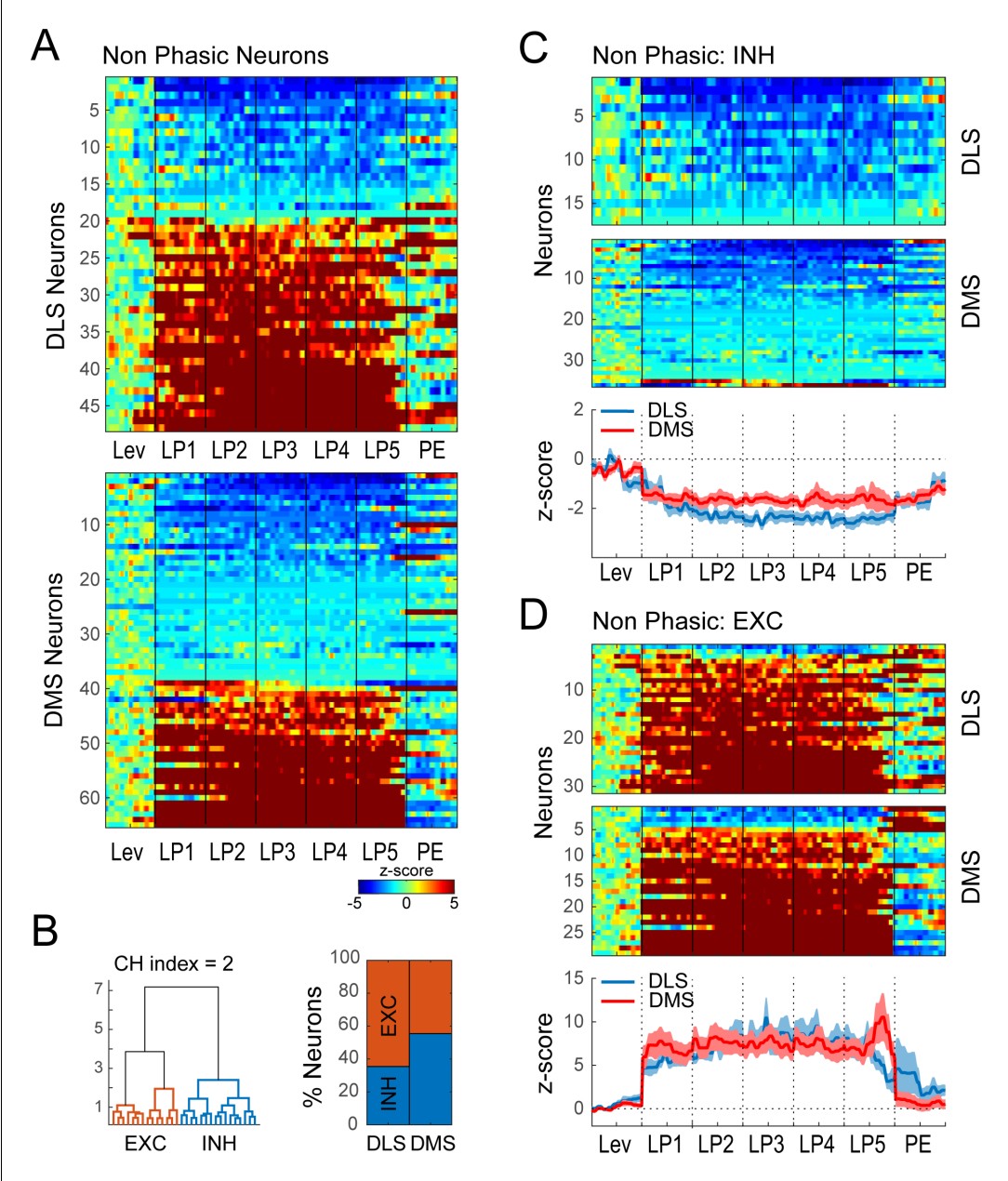

**Figure 5.** Hierarchical clustering on Non-phasic neurons during extended training. (**A**) Heatmap of normalized activity of neurons identified as being Non-phasic (48/338 neurons in DLS and 65/410 in DMS) sorted by averaged activity during the behavioral sequence. (**B**) Hierarchical clustering on Non-phasic neurons: dendrogram (left) and proportion of DLS and DMS neurons in the INH and EXC classes (right). (**C–D**) Heatmaps (top) and average z-score (± SEM) (bottom) of Non-phasic INH neurons (C; 17/48 and 36/65) and Non-phasic EXC neurons (D; 31/48 and 29/65) separated by regions; neurons are sorted by averaged normalized activity during the behavioral sequence.

DOI: https://doi.org/10.7554/eLife.49536.008

pattern, examination of the heatmaps illustrates overlap in activity patterns with some neurons falling under multiples categories. For instance some neurons combined start and stop activity (boundary neurons, *Figure 3C*), whereas others combine start inhibition and stop excitation (*Figure 3D*) or sustained inhibition and stop excitation (*Figure 5D*). Some neurons with detectable start and stop activity, but relatively weaker sustained activity, tended to be classified as Phasic Start or Phasic Stop, suggesting our approach for isolating sustained responses was relatively conservative.

## Large differences in relative proportions of Non-phasic firing patterns in DMS and DLS during early training

To determine the presence and relative proportions of the Phasic and Non-phasic classes in DMS and DLS during early training, we first separated Phasic and Non-phasic MSNs using the Fourier analysis, as described above (*Figure 6—figure supplement 1*). The proportion of Phasic and Non-phasic neurons in each session did not differ in DLS and DMS ($\chi^2$ <2.9, p-values>0.05; *Figure 6—figure supplement 1*). We then sought to classify the Phasic and Non-phasic neurons in the early training data set into different subtypes based on their activity, testing if the neuronal subtypes identified in the extended training dataset were represented in the early training dataset, and to what extent. To this end, we first trained a random forest classifier, a supervised learner, on the activity patterns of neurons from the extended dataset, along with their class labels obtained from the unsupervised hierarchical clustering. Then, we applied this classifier to the early training dataset to investigate the response type and their relative proportions in DMS and DLS across early training sessions.

The proportion of neurons in Phasic classes was similar between DLS and DMS for most sessions ($\chi^2$ <4.4, p-values>0.1), except the 4th session, where there was a greater fraction of Stop neurons and fewer Middle neurons in DLS (*Figure 6A*). On the other hand, neurons in Non-phasic classes were differentially distributed in DLS versus DMS in every session ($\chi^2$-values > 5.5, p-values<0.05; *Figure 6E*). More specifically, the majority (on average, 83%) of DLS Non-phasic neurons were classified as EXC whereas only an average of 21% of DMS non-phasic neurons were. DMS Non-phasic neurons were much more represented in the INH class (*Figure 6E–G*). Overall, these results show that DMS and DLS neurons express similar types of behavioral correlates during early training, but with large differences in their relative proportions. This over-representation of sustained excitation in the DLS, and of sustained inhibition in the DMS likely explains the regional differences observed in the analysis of the average activity during the first and last early training sessions (*Figure 2*).

Of note, while behavioral performance was improving over time, we were surprised to observe no obvious shifts in neural activity patterns from session 1 to session 10 in any individual class of Phasic or Non-phasic neurons. We found no significant change in class proportions across sessions (DLS, $\chi^2$ = 39, p>0.1; DMS, $\chi^2$ = 41, p>0.1). Analayzing the normalized activity of all units in a given class, we also obtained no evidence for changes across early training sessions (session F-values <1.4, p-values>0.1; session*event F-values <1.2, p-values>0.1; *Figure 6*). These ANOVAs however did detect regional differences in the magnitude of the mean normalized activity within the Start, Stop and Middle populations of neurons when comparing DMS and DLS (Start neurons: region $F_{(1,191)}$=91.3, p<0.0001; region*event: $F_{(11,2101)}$ = 20.08, p<0.0001; Stop neurons: region $F_{(1,112)}$=13.65, p<0.001; region*event $F_{(11,1232)}$ = 5.62, p<0.01; Middle neurons: region $F_{(1,433)}$=63.89, p<0.0001; region*event $F_{(11,4763)}$ = 9.71, p<0.0001; *Figure 6B–D*), reflecting higher average firing in DLS during lever presses, and higher activity after lever insertion and retraction in the DMS.

This classification analysis looking at individual neuronal responses confirms the findings based on average ensemble activity in showing that DMS and DLS activity differs during early training and is much more similar after extended training. This increase in similarity appears to mainly result from differences in sustained activity responses. After extended training, there is a higher proportion of INH response patterns in DLS and a higher proportion of EXC response patterns in DMS relative to early training leading to similar average responding.

## Neuronal activity during performance of the DT5 sequence task is sufficient for decoding striatal subregion identity during early training but not after extended training

An alternative approach to test the relative difference or similarity in DMS and DLS neural activity in early and extended training is to use that activity to decode the brain region identity of all recorded putative MSNs (*Figure 7* and *Figure 7—figure supplement 1*). Using ten-fold cross-validation, we determined how linear discriminant analysis (LDA) models trained on neural ensembles of normalized peri-event spike activity across the sequence from lever insertion to port entry (Materials and methods) could classify individual neurons as belonging to the DLS or DMS.

LDA models accurately revealed brain region identity during early training. More specifically, mean accuracy for assignment to DMS or DLS was significantly above chance for all early training

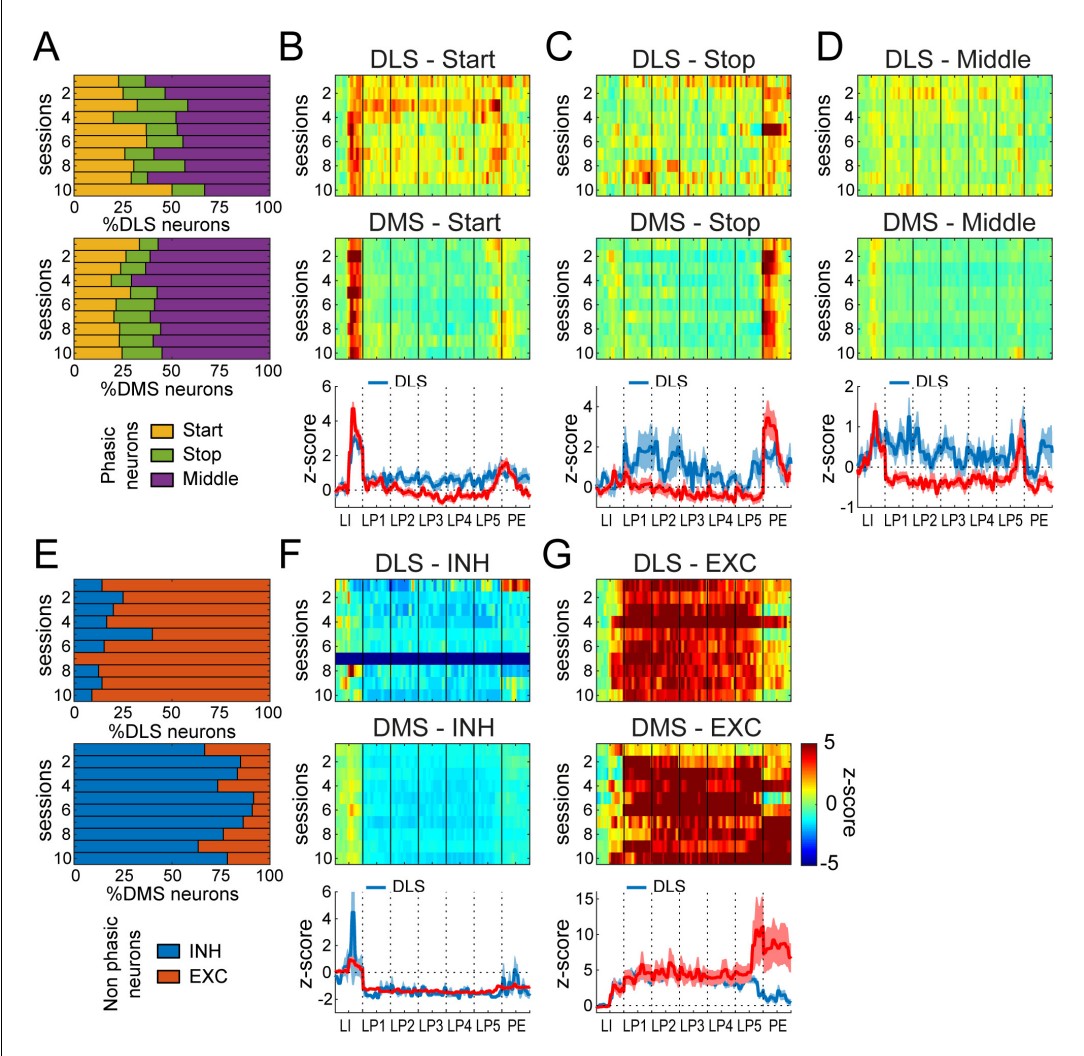

**Figure 6.** Differential neuronal responses in DMS and DLS during early training. (**A**) Proportion of Phasic Start, Stop and Middle neurons across training sessions. (**B–D**) Heatmaps of ensemble activity of Start (**B**), Stop (**C**), and Middle (**D**) neurons across training sessions in DLS (top) and DMS (middle) and average z-score (± SEM) across the last three sessions in DLS and DMS neurons (bottom). (**E**) Proportion of INH and EXC neurons across training sessions in DLS (top) and DMS (bottom). (**F–G**) Heatmaps of ensemble activity of INH (**F**) and EXC (**G**) neurons across training sessions in DLS (top) and DMS (middle) and average z-score (± SEM) across the last three sessions in DLS and DMS neurons (bottom). The absence of INH neurons in DLS on session 7 (**F**) is illustrated with a gray bar throughout the sequence.
DOI: https://doi.org/10.7554/eLife.49536.010

The following figure supplement is available for figure 6:

**Figure supplement 1.** Fourier analysis and hierarchical clustering allowing separation of Phasic and Non-phasic neurons during early training.
DOI: https://doi.org/10.7554/eLife.49536.011

sessions (p-values<0.0001, permutation test; *Figure 7A*). In contrast, for extended training, decoding accuracy did not differ from chance, even for the largest ensemble size (p-values>0.05, permutation test; *Figure 7A*). These results demonstrate that DMS and DLS neurons cannot be differentiated based on their sequence-related activity after extended training. These finding were confirmed using a lower-dimensional approach wherein a principal component analysis (PCA) was conducted on the concatenated z-scored neural activity around each event of the behavioral sequence (Materials and methods) and LDA models were trained using the first three principal components obtained from each early training session and from extended training. This analysis produced similar results, with decoding accuracy significantly above chance in all early training sessions

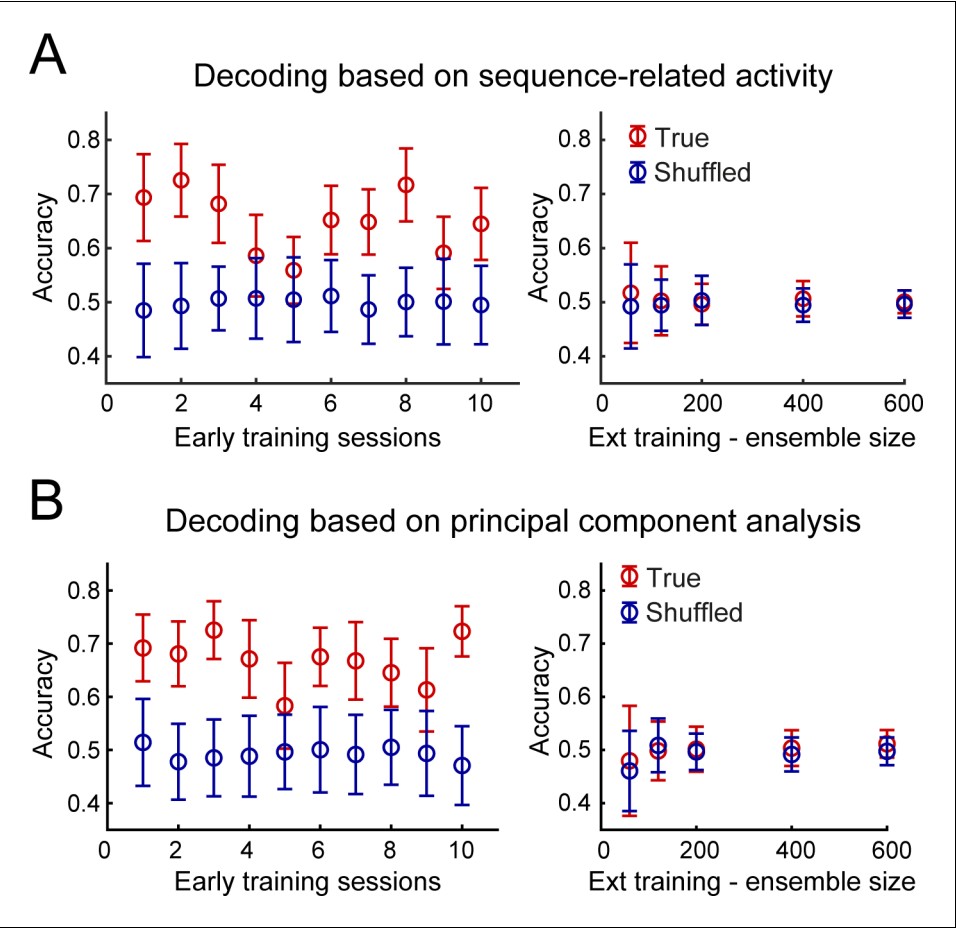

**Figure 7.** Accurate decoding of the dorsostriatal brain region identity in early training but not after extended training. (**A**) Mean decoding accuracy (± standard deviation) of the dorsostriatal subregion identity, DLS or DMS, of individual neurons based on their sequence-related activity (True - red) is compared with the decoding accuracy expected from chance (Shuffled - dark blue) across early training sessions (left) or during extended training, as a function of ensemble size (right). (**B**) Decoding based on the first three principal components following principal component analysis. Average accuracy (± standard deviation) is presented as a function of early training sessions (left) or ensemble size during extended training (right).

DOI: https://doi.org/10.7554/eLife.49536.012

The following figure supplement is available for figure 7:

**Figure supplement 1.** Principal Component Analysis (PCA) followed by Linear Discriminant Analysis (LDA) allows prediction of brain region identity in early training but not extended training.

DOI: https://doi.org/10.7554/eLife.49536.013

---

(all p-values<0.0001, permutation test) but not for extended training, even for larger sets of units (all p-values>0.05, permutation test; *Figure 7B*).

These findings are consistent with the differences in proportions of neuron subtypes across the two regions in early training; this difference in proportions permits a reliable inference of which region a neuron comes from when using an unbiased decoding strategy. Altogether, these findings demonstrate through three approaches that DMS and DLS activity was relatively dissimilar during early training but similar after extended training.

## Individual differences in sensitivity to outcome devaluation do not substantially correlate with dorsostriatal activity

Although, on average and as a group, lever pressing performance was insensitive to reward devaluation after early and extended training, we observed inter-individual variability in the sensitivity to

outcome devaluation (*Figure 1H-I*). Thus, we sought to determine whether DMS and DLS activity differs across early training and extended training in subjects relatively insensitive or sensitive to devaluation, considering this as a possible read-out of the degree of habitual versus goal-directed control (*Figure 8*). Rats with a devaluation ratio above 0.45 were considered as insensitive to outcome devaluation (*Figure 8A–B*). Examining the average normalized activity of DMS and DLS neurons along the behavioral sequence across early training sessions (*Figure 8A*), we observed significant region by group and region by group by event interactions during early training (region*group: $F_{(1,819)}$=22.36, p<0.0001; region*group*event: $F_{(11,9009)}$ = 4.3, p<0.01), but no main effect of group ($F_{(1,819)}$=0.69,p>0.4) nor event by group interaction ($F_{(11,9009)}$ = 1.24,p>0.2). When separated into individual classes of neurons (*Figure 8C*; *Figure 8D*), the neuron numbers are too small for meaningful analysis of relative proportions or magnitudes of responses; however the percentages within each class look similar (*Figure 8—figure supplement 1A*), while observation of the heatmaps suggest relatively greater activity in the DLS of sensitive subjects. It is unclear from these results whether specific activity patterns emerge across early training sessions in rats whose responding transfers toward habitual control.

When looking at the normalized activity in DMS and DLS in rats more or less sensitive to outcome devaluation during extended training, we found no significant differences (*Figure 8B*; main effect of group: $F_{(1,568)}$=2.84, p>0.05; group*region interaction: $F_{(1,568)}$=1.32, p>0.1; group*event interaction: $F_{(11,6248)}$ = 2.17, p>0.05; group*event*region interaction: $F_{(11,6248)}$ = 1.67, p>0.1). The activity and distribution of DLS and DMS neurons was overall similar across classes of neurons in the two groups of subjects (*Figure 8E-F*; *Figure 8—figure supplement 1B*; DLS: $\chi^2$ = 8.81, p=0.066; DMS: $\chi^2$ = 6.78, p>0.1). Overall, there are no obvious correlations between degree of sensitivity to outcome devaluation and the neural activity patterns.

We have previously shown in non-tethered rats that sensitivity to reward devaluation by satiety was promoted in the DT5 task when rats are trained with liquid sucrose but not with grain-based pellets (*Vandaele et al., 2017*). Here, we have included data from 5 of 17 rats who were trained with pellet rewards. However, dividing the groups based on reward type did not reveal substantial differences in behavior and neuronal activity (*Figure 8—figure supplement 2*). We also note that rats trained with both sucrose and pellets reliably show insensitivity to reward devalution by conditioned taste aversion (*Vandaele et al., 2017*), supporting the notion that on average the DT5 procedure promotes habitual responding.

## Pharmacological inactivation of both DLS and DMS interferes with task performance

Although dorsostriatal activity did not correlate with habitual learning, we have found neural correlates of performance in the DT5 task in both DLS and DMS during early and extended training. To assess causal involvement of these regions in DT5 performance, additional groups of rats were bilaterally implanted with cannulae targeting DMS (N = 8) or DLS (N = 11) and trained to asymptotic performance (20 and 10 sessions, respectively). Rats received micro-infusions of saline or a cocktail of GABA agonists, baclofen and muscimol, before a DT5 session (*Figure 9A–B*). Inactivation of DMS resulted in a subtle but significant decrease in the number of lever presses (*Figure 9C*; $F_{(1,7)}$=7.36, p<0.05). We also observed a trend toward an increase in latency to the first lever press following DMS inactivation (*Figure 9D*; $F_{(1,7)}$=5.16, p=0.057), but no effect on response rate (*Figure 9E*; $F_{(1,7)}$=1.43, p>0.1). DLS inactivation slightly decreased the number of lever presses, although the effect was not significant (*Figure 9F*; Sign test: $Z_{11}$ = 1.8, p=0.07). We also observed a decrease in response rate following DLS inactivation (*Figure 9H*; $F_{(1,10)}$=5.51, p<0.05), but no effect on the first lever press latency (*Figure 9G*; Sign test: $Z_{11}$ = 0.6, p>0.5). The effect on response rate may result from a shift in the distribution of inter-press intervals to the right following DLS inactivation (*Figure 9J*; K-S test: $D_n$ = 0.17, p<0.0001) but not DMS inactivation (*Figure 9I*; K-S test: $D_n$ = 0.04, p>0.1). These results confirm the concurrent involvement of both DMS and DLS during sequence performance in the DT5-task.

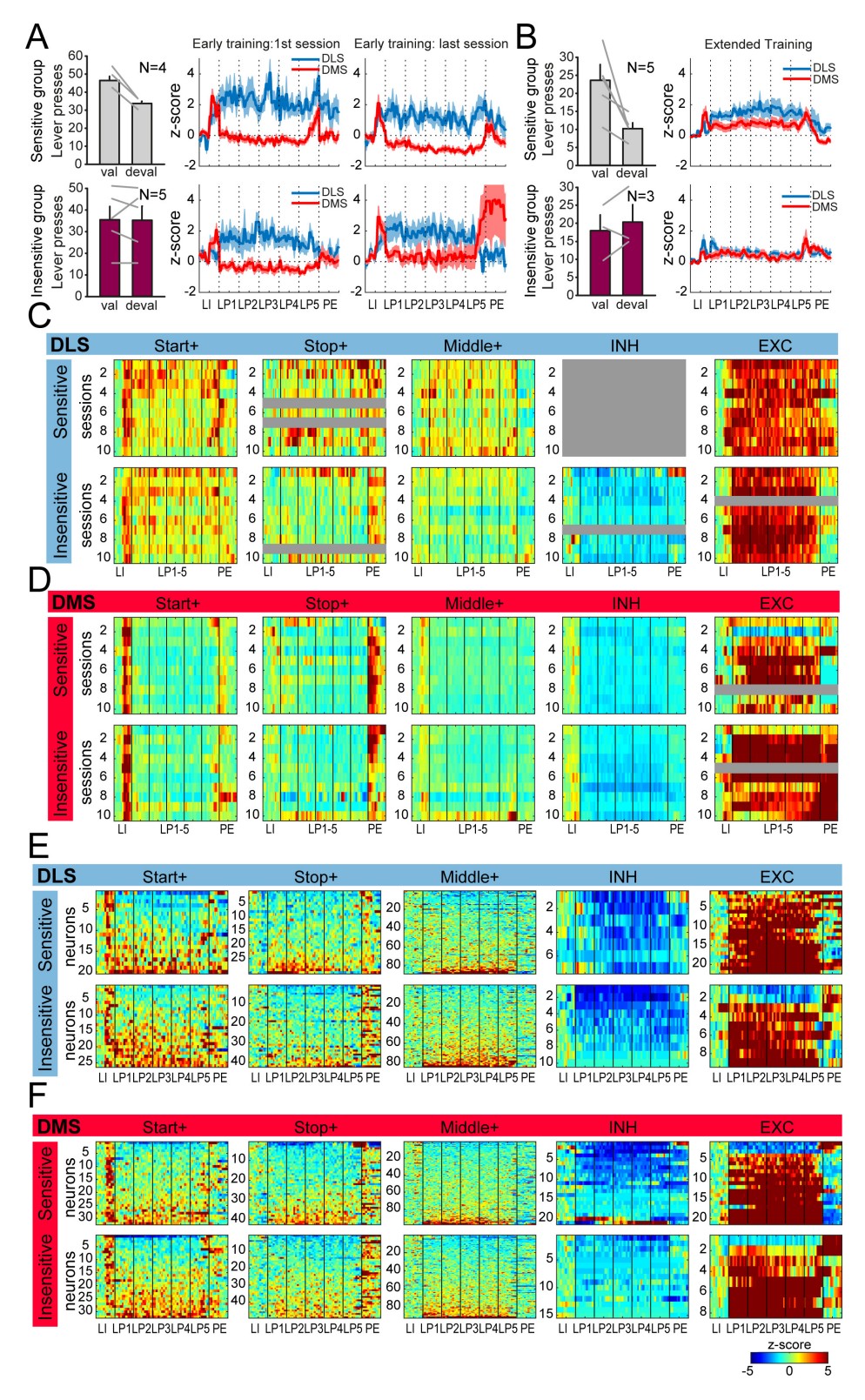

**Figure 8.** Similar dorsostriatal sequence-related activity in groups of rats differing in their sensitivity to satiety-induced devaluation. **(A)** Number of lever presses in the valued (val) and devalued (deval) conditions in rats sensitive (devaluation ratio <0.45) (top) or insensitive (devaluation ratio >0.45) (bottom) to satiety induced devaluation after early training (left), and average z-score (± SEM) in DLS and DMS neurons during the first (middle) and last (right) early training sessions in these 2 subgroups of rats (top: sensitive group; bottom: insensitive group). **(B)** Number of lever presses in the valued (val) and

*Figure 8 continued on next page*

*Figure 8 continued*

devalued (deval) conditions in rats sensitive (devaluation ratio <0.45) (top) or insensitive (devaluation ratio >0.45) (bottom) to satiety induced devaluation after extended training (left), and average z-score (± SEM) in DLS and DMS neurons (right) in these 2 subgroups of rats (top: sensitive group; bottom: insensitive group). (C–D) Heatmaps of ensemble sequence-related activity in DLS (C) and DMS (D) across early training sessions, in each class of neuronal responses (from left to right) and as a function of sensitivity to outcome devaluation (top: sensitive group; bottom: insensitive group). (E–F) Heatmaps of sequence-related activity of DLS (E) and DMS neurons (F) during extended training, in each class of neuronal responses (from left to right) and as a function of sensitivity to outcome devaluation (top: sensitive group; bottom: insensitive group). The absence of neurons in some classes on a given early training session is illustrated with a gray bar throughout the sequence.

DOI: https://doi.org/10.7554/eLife.49536.014

The following figure supplements are available for figure 8:

**Figure supplement 1.** Distribution of DMS and DLS neural activity patterns in rats sensitive or insensitive to satiety induced devaluation.
DOI: https://doi.org/10.7554/eLife.49536.015
**Figure supplement 2.** Similar behavior and dorsostriatal sequence-related activity in groups of rats trained with liquid sucrose and grain-based pellets.
DOI: https://doi.org/10.7554/eLife.49536.016

## Both DLS and DMS remain correlated with over-trained task performance during extended training

If the DLS is specifically involved in over-trained habitual/skilled performance when no additional learning or deliberation is required, then one would expect a disengagement of DMS from behavioral control after extended training. However, our findings show multiple behavior-related neural activity patterns in DMS at this time point. To further test this hypothesis, we looked at correlations between DMS and DLS neural activity and specific phases of the behavior to ask whether these regions were differentially correlated with over-trained performance. We examined the trial-by-trial relation between (i) neural activity at lever insertion and latency to the first lever press, (ii) neural activity during the lever press sequence and response rate, and (iii) neural activity following lever retraction and latency to port entry (*Figure 10A–C*, *Figure 10—figure supplement 1*). We controlled for significant correlations occurring spuriously in a small percentage of units by running spearman correlations on 1000 iterations of shuffled data for each behavioral variable (*Figure 10— figure supplements 2–4*). The percentage of correlated units in each region significantly deviated from the proportion expected by chance for each behavioral variable except correlations between DLS neural activity and port entry latency (*Figure 10—figure supplement 4*, p=0.1). We found that spike activity of 42% of recorded MSNs in DLS and DMS was correlated with performance (*Figure 10D*), with significant regional differences ($\chi^2$=9.4, p<0.05; *Figure 10E*). There was no significant difference between DMS and DLS in the proportion of neurons correlated with latency to $1^{st}$ lever press (DLS: 16.6%, N = 56; DMS: 18.8%, N = 77) or response rate (DLS: 14.8%, N = 50; DMS: 13.7%, N = 56; *Figure 10E*). However, the proportion of neurons correlated with port entry latency was surprisingly higher in DMS than DLS (DLS: 6.8%, N = 23; DMS: 11.9%, N = 49; $\chi^2$ = 5.6, p<0.05; *Figure 10E*). Thus, DMS activity remains correlated with optimized performance, and appears in fact more involved at the termination of the behavioral sequence than DLS. There was overall a small fraction of neurons with multiple behavioral correlations (*Figure 10D*). Correlated neurons were represented in every class of neuronal response determined above (*Figure 10—figure supplement 1*), suggesting that each of the sequence-related activity patterns isolated in this study may contribute to ongoing performance in the DT5 task.

## Discussion

Here, we characterized and compared DLS and DMS sequence-related activity during improvement in skilled performance across early training sessions and during extended training, likely after considerable skill consolidation. Neural activity in DMS and DLS was strongly dissimilar during early training and did not show obvious changes across sessions despite significant improvement in performance. However, DMS and DLS activity was more similar after extended training, with a comparable distribution of sequence-related behavioral correlates in the neuronal activity across the two regions. Optimized performance after extended training was not associated with DMS disengagement from behavioral control, since both DMS and DLS neurons were involved in initiation, execution and termination of the lever pressing sequence, as indicated by trial-by-trial correlations between neural firing

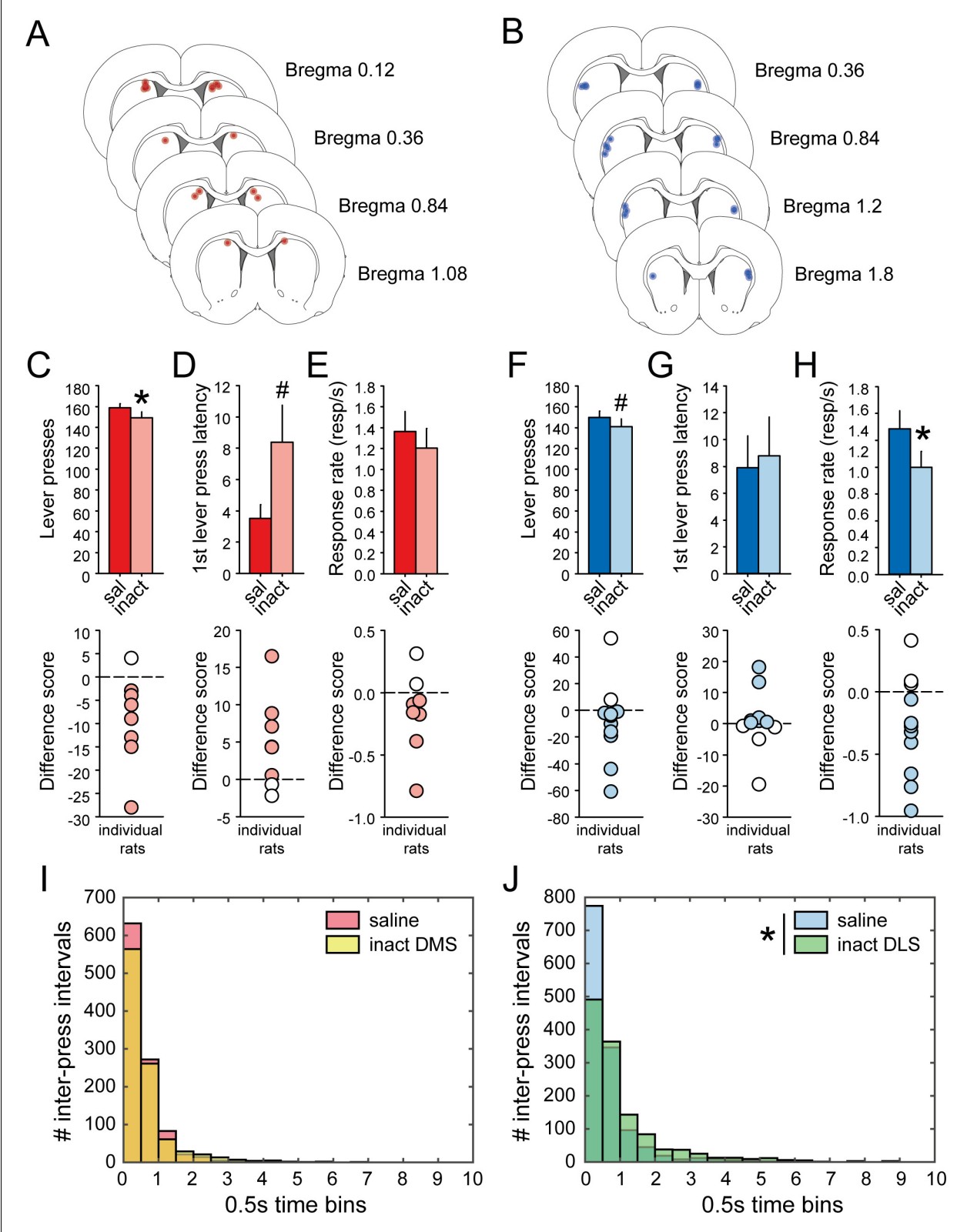

**Figure 9.** Pharmacological inactivation of DMS and DLS disrupts task-performance. (**A–B**) Schematic representation of cannula placements in DMS (**A**) or DLS (**B**). (**C–E**) Mean number of lever presses (± SEM) (**C**) mean first lever press latency in seconds (± SEM) (**D**) and mean response rate (± SEM) (**E**) after infusion of saline or muscimol/baclofen in the DMS (top) and difference scores (inactivation – saline) of individual rats (bottom). (**F–H**) Mean number of lever presses (± SEM) (**F**), mean first lever press latency in seconds (± SEM) (**G**), and mean response rate (± SEM) (**H**) after infusion of saline or

*Figure 9 continued on next page*

*Figure 9 continued*

muscimol/baclofen in the DLS (top) and difference scores (inactivation – saline) of individual rats (bottom). (I–J) Distribution of inter-press intervals after infusion of saline (DMS, red; DLS, blue) or muscimol/baclofen (DMS, yellow; DLS, green) in the DMS (I) or in the DLS (J).

DOI: https://doi.org/10.7554/eLife.49536.017

and behavior. Furthermore pharmacological inactivation of both DLS and DMS had moderate effects on performance. These results suggest that behavioral sequences triggered by salient stimuli may continue to involve both the DMS and DLS long after initial acquisition, even when performance is highly regular.

We previously showed that responding for sucrose reward on a free-running fixed-ratio five schedule remains goal directed even after 43 training sessions, while, in contrast, rats trained on the discrete-trials fixed-ratio 5 (DT5) task used here are relatively insensitive to devaluation as early as the 5th training session (*Vandaele et al., 2017*). In addition, rats trained under the DT5 procedure show lower trial-by-trial variability in responding than rats trained under free-running fixed-ratio 5, in agreement with greater regularity and automaticity in DT5 trained subjects (*Vandaele et al., 2017*). We hypothesized that the rapid development of habitual and automatic responding in rats trained under the DT5 schedule is due to the nature of lever presentation, in which insertion acts as a cue to begin lever pressing, and lever retraction signals the end of responding. Thus sequence initiation may be triggered by lever insertion, and lever retraction can ameliorate any requirement for monitoring the number of lever press responses. We therefore expected to observe strong sequence-related activity in the DLS, in accordance with prior findings (*Barnes et al., 2005*; *Jin and Costa, 2010*; *Jin et al., 2014*; *Martiros et al., 2018*; *Thorn et al., 2010*). In addition, we expected DMS behavior-related activity to decrease over time as behavioral performance presumably depended increasingly on the DLS. Our findings did not fully support our expectations: we observed neural activity correlated with the behavioral sequence but did not find evidence to support a disengagement of the DMS over time.

Using standard single-neuron analyses, we found units that either increased or decreased their activity in association with one or more of the behavioral events we measured. This approach revealed differences in the relative proportions of excitations and inhibitions in DMS and DLS in early training, but did not help us distinguish the characteristic firing patterns we observed. We therefore turned to a classification approach to identify behavioral correlates in the spiking patterns of DMS and DLS neurons. This data-driven approach revealed many of the neural response types previously described during lever pressing sequences (*Jin et al., 2014*). We found subpopulations of Phasic neurons showing transient excitation at the start or at the end of the sequence, whereas Non-phasic neurons were characterized by sustained excitation or inhibition during the sequence. Both sustained excitation throughout sequentially-performed actions and excitation marking the beginning and termination of sequentially-performed actions have been proposed to represent the chunking of individual actions into a single unit (*Barnes et al., 2005*; *Graybiel and Grafton, 2015*; *Jin and Costa, 2015*; *Jin and Costa, 2010*; *Jin et al., 2014*; *Smith and Graybiel, 2016*), while excitations at the boundaries of sequences have been hypothesized to signal the initiation and termination of the sequence (*Jin et al., 2014*; *Jin and Costa, 2010*; *Jin and Costa, 2015*; but see *Sales-Carbonell et al., 2018*). As would be predicted by prior findings (e.g., *Thorn et al., 2010*), the distribution of these neural response patterns was indeed different in DMS and DLS during early training. Specifically, in the DLS, activity was characterized by a large proportion of neurons showing sustained excitation during the entirety of the series of lever presses, with a relatively low proportion of neurons expressing sustained inhibition throughout. A somewhat opposite pattern was observed in DMS with excitation mostly restricted to the boundaries or delimiters of the sequence in Phasic neurons (i.e, at lever insertion and lever retraction), and more neurons expressing sustained inhibition during lever pressing. However, the activity in DMS did not diminish over time, and DLS and DMS neural activity in fact became more similar after months of overtraining.

The presence of task-bracketing activity in DMS was unexpected. Previous work has demonstrated the development of task-bracketing activity in DLS, but not in DMS, as rats learned to navigate in a T-maze (*Barnes et al., 2005*; *Regier et al., 2015*; *Smith and Graybiel, 2013*; *Thorn et al., 2010*). Indeed, in the study of Thorn and colleagues, DLS activity was characterized by excitation at

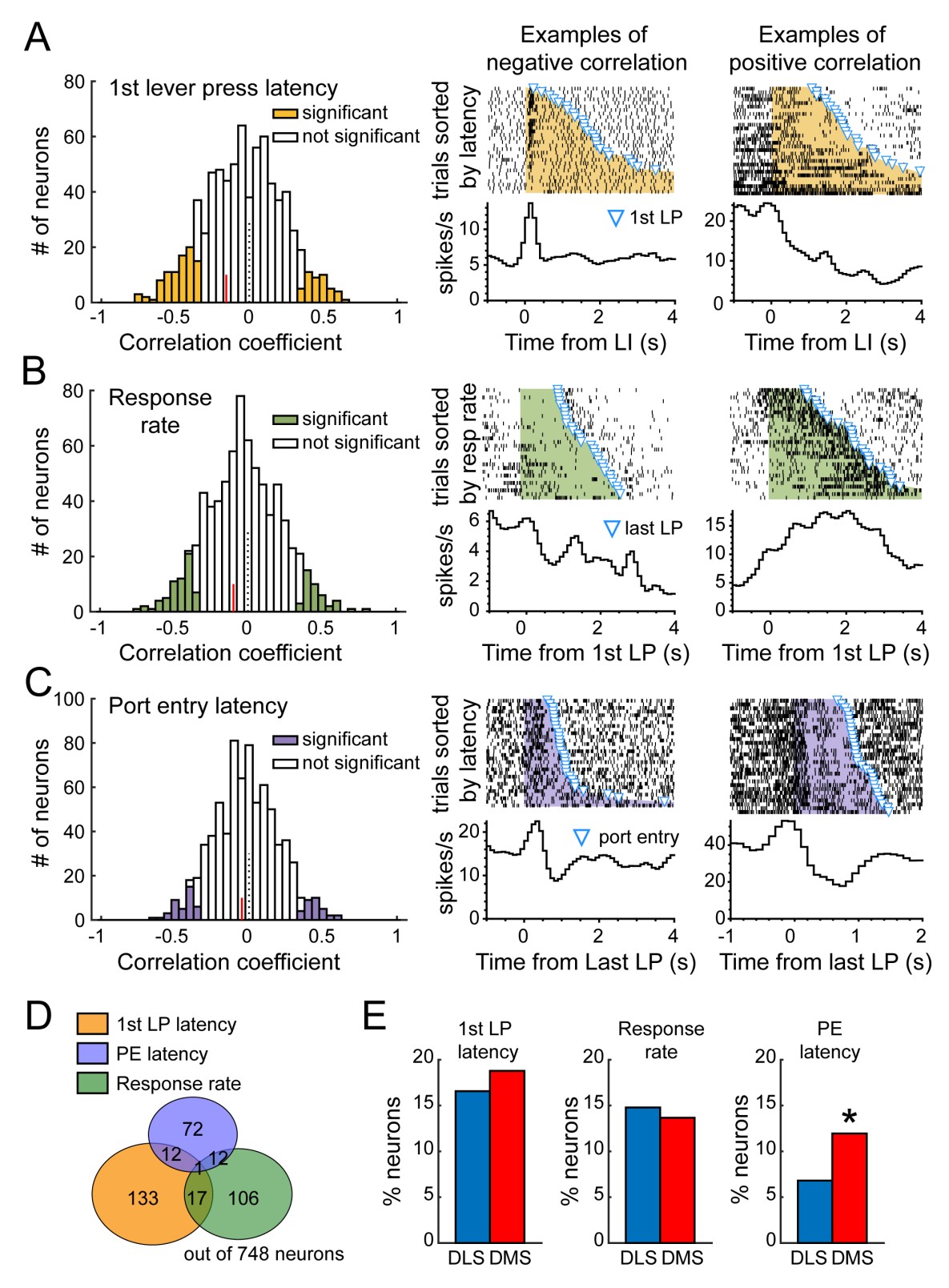

**Figure 10.** DLS and DMS neural activity correlates with over-trained task-performance during extended training. (**A**) Left: Distribution of Spearman rank correlation coefficients relating firing rate after the lever insertion (0–500 ms) to the first lever press latency on a trial by trial basis. Example of neurons with a negative (middle) or positive correlation (right) between firing rate and latency. (**B**) Left: Distribution of Spearman rank correlation coefficients relating firing rate during the sequence (from first to last lever press) to the response rate on a trial by trial basis. Example of neurons with a negative

*Figure 10 continued on next page*

*Figure 10 continued*

(middle) or positive correlation (right) between firing rate and response rate. (**C**) Left: Distribution of Spearman rank correlation coefficients relating firing rate after the lever retraction (0–500 ms) to the port entry latency on a trial by trial basis. Example of neurons with a negative (middle) or positive correlation (right) between firing rate and port entry latency. Red lines on correlation coefficient histograms indicate the coefficient median of units significantly correlated. (**D**) Venn diagram illustrating the number of units with significant correlation between firing rate and 1st lever press latency (yellow), response rate (green) and port entry latency (blue). (**E**) Proportion of neurons in DMS and DLS expressing correlation with the 1st lever press latency (left), the response rate (middle), and the port entry latency (right).

DOI: https://doi.org/10.7554/eLife.49536.018

The following figure supplements are available for figure 10:

**Figure supplement 1.** Comparison of correlations relating neural firing rate and behavior in DMS and DLS.

DOI: https://doi.org/10.7554/eLife.49536.019

**Figure supplement 2.** Characterization of correlations relating firing rate post-lever insertion and 1st lever press latency.

DOI: https://doi.org/10.7554/eLife.49536.020

**Figure supplement 3.** Characterization of correlations relating within-sequence firing rate and response rate.

DOI: https://doi.org/10.7554/eLife.49536.021

**Figure supplement 4.** Characterization of correlations relating firing rate post-lever retraction and port entry latency.

DOI: https://doi.org/10.7554/eLife.49536.022

the boundaries of the navigation sequence whereas DMS activity peaked in the middle, when animals chose between two directions based on instruction cues. A more recent study by *Martiros et al. (2018)* also reported neuronal excitation at the beginning and end of a 3-response lever press sequence in the DLS, but not in the DMS. In contrast, in the present study, DMS activity peaked at the lever insertion and lever retraction cues and was inhibited during lever pressing, whereas DLS neurons expressed sustained excitation throughout the sequence. One possible explanation for these differences is that both the T-maze task and the 3-response sequence task require the subject to move about the apparatus, and thus to link actions across space. In contrast, in the current DT5 task, rats responded repetitively upon a single operandum in the same location. In agreement with this notion, both sustained excitation and inhibition were observed in the dorsal striatum of mice similarly performing a series of lever presses (*Jin and Costa, 2010*; *Jin et al., 2014*). Sustained excitation in DLS neurons is also consistent with studies recording DLS activity in tasks involving locomotor sequences (*Rueda-Orozco and Robbe, 2015*; *Sales-Carbonell et al., 2018*).

A second possibility for why task-bracketing activity was strongly observed in the DMS is that these responses may largely represent cue-elicited activations rather than motor sequence initiation-cessation signals. Although we isolated neurons showing activity consistent with a start-stop characterization, a major distinction between the present findings and much of the prior work is that the beginning and end of the behavioral sequence in this study is cued by lever insertion and retraction. These two events proved to be strong stimuli that elicited excitatory neural activity resembling previously-described start/stop or boundary neurons (*Jin et al., 2014*; *Smith and Graybiel, 2013*). Yet close examination revealed that most so-called 'Start neurons' responding to the lever cue at trial initiation did not respond just before the first lever press. These data suggest the alternative explanation that phasic activity associated with behavioral sequence initiation, including that observed here, may reflect sensory cues rather than strictly the initiation of a series of actions. This result is consistent with prior findings showing an absence of start signals at the initiation of a locomotor sequence (*Sales-Carbonell et al., 2018*). Further, Phasic Stop neurons expressed excitation at the termination of the sequence after the lever retraction cue but also before the port entry as rats approached the magazine to retrieve the reward. This excitation was not found before isolated port entries during inter-trial intervals and may therefore encode an expectation of the reward, triggered by the lever retraction cue.

Based on our previous study, we expected a transition from goal-directed to habitual control over early training sessions (*Vandaele et al., 2017*). Yet, neural activity was relatively similar across the first ten sessions of DT5 training, even as markers of behavioral regularity and automaticity improved. Thus, we cannot conclude that the neural correlates we report here mediate the development of habit or the improvement in performance over time. Of note, as a group, subjects in the early training group already showed habitual responding after 10 sessions. However, our analysis suggests that the specific DMS and DLS neural patterns observed during those 10 sessions do not

substantially correlate with the presence or absence of habitual control, because when we separated subjects based on habitual or goal-directed control for early (or extended) training, the neural patterns were largely similar. This conclusion, however, rests on our arbitrary separation of rats by sensitivity to outcome devaluation by satiety, which may not be the best metric for assessing habitual control, since evidence with other manipulations suggests strong habit formation in the DT5 task (*Vandaele et al., 2017*). Overall, it is not clear whether or how sequence-related activity relates to either skill learning/performance or habit learning/performance. Future studies could address this question by comparing dorsostriatal activity in the DT5 task with free-running instrumental behavior that remains goal directed after extended training (*Vandaele et al., 2017*).

We focused on neural activity at the beginning of DT5 training, but all subjects experienced prior instrumental training under CRF and DT1 during which relevant aspects of the behavior and neural activity may have become organized. Although within-sequence response rate progressively increased across the ten sessions, the number of within-sequence port entries, a marker of behavioral chunking, decreased quickly with most of the change occurring in session one, suggesting rapid concatenation of individual lever presses into unitary sequences (*Figure 1D*, inset). In fact, in the DT5 task, rats do not have to track the number of presses emitted to obtain the reward and may just continue pressing on the lever until its retraction, which may favor behavioral chunking and the specific activity patterns reported in this study. Thus, rapid acquisition of the new response requirement, facilitated by the presence of cues predicting reward availability and delivery, could explain why sequence-related activity is observed in both regions very rapidly, compared to previous studies using uncued, self-paced procedures or requiring discriminatory learning (*Barnes et al., 2005*; *Jin and Costa, 2010*; *Jin et al., 2014*; *Smith and Graybiel, 2013*; *Thorn et al., 2010*). Including probe trials in which lever retraction is delayed would be helpful for to determining to what extent behavior and neural activity depend on the lever retraction cue.

Skill learning is generally characterized by an initial phase of rapid performance improvement followed by an extended period of slow performance optimization (skill consolidation). We observed that during extended training, execution of lever press sequences was optimized and stereotyped, with within-sequence response rate, trial-by-trial variability in response rate, and $1^{st}$ lever press latency reaching asymptotic levels. We hypothesized that DLS sequence-related activity would strengthen with repeated practice, whereas DMS, involved in initial goal-directed learning, would disengage, thereby increasing the disparity between these two regions. Contrary to our predictions, correlations with behavior suggest that DMS did not disengage from behavioral control after extended training (see below), and neuronal activity in DMS and DLS became more similar. Ensemble activity analysis revealed that, on average, both DMS and DLS expressed sustained excitation throughout the lever press sequence with peaks following lever insertion and retraction. While there was a moderate difference in the relative proportions of sustained inhibitory activity in the two regions, DMS and DLS neurons could not be differentiated by LDA models based on their sequence-related activity, a result that stands in sharp contrast with the dissociation between these brain regions during early training. The increase in dorsostriatal similarity after extended training resulted from an increase in sustained inhibition in DLS and a higher proportion of sustained excitation neurons in DMS relative to early training. Furthermore, we observed less activity during lever presses and stronger activity in response to lever insertion and retraction in Phasic DLS neurons after extended training compared to early training (compare *Figure 3C–D*). One interpretation is that these changes reflect greater cue encoding in DLS and greater action encoding in DMS. This suggests that skill consolidation following many weeks of training in this task leads to improved integration of both cues and actions across regions within the dorsal striatum. Few studies have examined neuronal activity in dorsal striatum across two training stages so separated in time, so it is not yet clear whether findings will be similar after extended training (months) in other tasks.

If the neural activity as characterized here does not clearly correlate with habit or automatization, what might these patterns represent? As mentioned above, the relative absence of change across early training may reflect rapid behavioral chunking promoted by the lever insertion and retraction cues. While the proportions of distinct classes of neuronal activity patterns that might represent chunking were different in DLS and DMS, both regions do show these types of patterns, and these patterns were similar after extended training. Yet, the functional contribution of these DMS and DLS signals to behavior is expected to be distinct based on the specific connectivity of these regions as nodes within parallel corticostriatal loops. After extended training, a greater proportion of DMS

than DLS neurons showed significant trial-by-trial correlations between neuronal activity after the final lever press and latency to enter the reward port. This suggests that DMS neural activity may mediate the impact of associations between the lever retraction stimulus or the port entry action with the rewarding outcome (*Burton et al., 2015*; *Ito and Doya, 2015*; *Kimchi and Laubach, 2009b*; *Kimchi and Laubach, 2009a*) even after substantial training. Interestingly, similar proportions of neurons in DMS and DLS expressed correlations with the 1st lever press latency and the response rate, suggesting that both regions are involved at the initiation and execution of the sequence. In fact, pharmacological inactivation of both DLS and DMS modestly reduced responding during the task, although through different mechanisms. These results are in agreement with our correlation analysis and prior findings in showing concurrent engagement of both DLS and DMS in performance vigor (*Kim et al., 2014*; *Panigrahi et al., 2015*; *Rueda-Orozco and Robbe, 2015*). Taken together, the correlation analyses and the inactivations support a role for sensorimotor striatal activity in moment-by-moment expression of behavior (*Robbe, 2018*; *Rueda-Orozco and Robbe, 2015*; *Sales-Carbonell et al., 2018*), suggesting that the neural representation of higher-order functions, including transitions in cognitive control, may be more clearly reflected in activity in other regions, or in broader neuronal loops that include the striatum.

To conclude, we observed dissociated DMS and DLS sequence-related activity in early training that did not evolve across early training sessions, despite significant changes in performance. The biggest change in dorsostriatal sequence-related activity occurred between early and extended training, two training stages involving very moderate changes in behavior in our experimental conditions. Thus, although the activity of 41% of neurons was correlated with performance on a trial-by-trial basis, changes in sequence-related activity within and across training stages appeared to be uncoupled with changes in behavior. Furthermore, although behavioral control was on average habitual in both early training and extended training groups, individual differences in sensitivity to outcome devaluation did not correlate well with dorsostriatal activity. Thus, it is not clear how activity in DMS and DLS relates to skill learning and formation of habit. One interpretation is that the absence of change across early training reflects rapid behavioral chunking promoted by the lever insertion and retraction cues. Furthermore, the alterations in sequence-related activity over extended training may reflect a refinement of the temporal properties of striatal circuit engagement that may not map one-to-one with the behavioral variables we and others typically measure. It is possible that temporally-correlated circuit engagement across parallel corticostriatal loops is associated with better performance and/or skill consolidation, but this remains to be tested. Together, these findings suggest that sequence-related activity in the dorsal striatum during early and extended training may reflect additional processes not captured by our behavioral indicators of sequence learning, optimization of performance, or habits.

## Materials and methods

### Subjects

#### Subjects

Male Long Evans rats (N = 36, Harlan, IN) were individually housed and maintained in a light- (12 hr light-dark cycle, lights on at 7am) and temperature-controlled vivarium (21° C). Except just before and after surgery, rats were maintained at 90% of their free-feeding weight; food rations were given 1–2 hr after daily behavioral sessions. Water was available ad libitum. This study was carried out in accordance with the recommendations of the Guide for the Care and Use of Laboratory Animals (National Research Council, 1996), and was approved by the institutional animal care and use committee of Johns Hopkins University.

#### Experimental groups

Following instrumental training, rats in the early training group (N = 9) underwent surgery, and, after recovery, neural activity in dorsal striatum was recorded throughout acquisition of the DT5 task, which included 3 DT1 sessions and 10 DT5 sessions. Rats in the extended training group (N = 8) received extensive training in the DT5 task with more than 45 DT5 sessions prior to surgery; neural activity in these subjects was recorded daily during an additional 9 to 22 DT5 sessions (total training:

58–67 DT5 sessions). Sensitivity to satiety-induced devaluation was assessed at the end of recording for all rats.

Pharmacological inactivation of DMS (N = 8) or DLS (N = 11) were tested on two additional groups of rats, trained in the DT5 task for 20 and 10 sessions, respectively. Effect sizes were not estimated before the experiment but we aimed at including a minimum of 8 rats per group with correct cannulae placements based on established standards.

## Behavioral training

### Behavioral apparatus and DT5 task training

Training occurred in conditioning chambers designed for in vivo neural recording housed within sound-attenuating boxes (Med Associates, St Albans, VT). The house light, located on the ceiling of the chamber, remained illuminated during the full length of every session. DT5 training was conducted as described in *Vandaele et al. (2017)*. Rats first underwent a single 30 min magazine training session, in which reward was delivered under a variable time-60s schedule. Rats were next trained to press the left lever to earn reward, delivered in the adjacent magazine. Rats were trained to respond for either 20% sucrose (0.1 mL delivered over 3 s) (early training: N = 6, extended training: N = 6) or a single grain-based pellet (Bioserv Biotechnology) (early training: N = 3, extended training: N = 2). Sessions were limited to 1 hr or 30 reward deliveries. Next, subjects advanced to discrete trial (DT) training, in which each session consisted of 30 trials separated by 1 min inter-trial intervals. Every trial was initiated by insertion of the left lever. For the first three sessions, one lever press simultaneously resulted in the retraction of the lever, reward delivery, and initiation of a new inter-trial interval (discrete-trial fixed ratio (FR) 1; DT1). On session 4, the response ratio was increased to 5 (discrete-trial FR5; DT5). Failure to complete the ratio within 1 min was considered as an omission and resulted in lever retraction and initiation of a new inter-trial interval.

### Outcome devaluation by sensory-specific satiety

To avoid neophobia, rats were pre-exposed to the control reward for 30 min in feeding cages one or two days before the first devaluation test. Each rat received 2 days of testing, separated by one reinforced training session. On the first test day, half of the rats were given free access to their training reward (devalued condition), while the other half received a control reward, which never served as a reinforcer (valued condition). Grain-based pellets served as the control reward for rats trained with liquid sucrose, and liquid sucrose served as the control reward for rats trained with pellets. Pre-feeding occurred for 1 hr in feeding cages in the experimental room. Immediately after pre-feeding, rats were tethered and placed in the recording chambers for a 10-trial extinction session. The procedure was identical to that of training except that no reward was delivered. On the second test session, animals were pre-fed with the alternative reward prior to the 10-trial extinction test.

### Surgery

Rats underwent surgery under isoflurane anesthesia (0.5–5%), after receiving pre-operative injections of cefazolin (75 mg/kg, antibiotic) and carprofen (5 mg/kg, analgesic). Topical lidocaine was applied for local analgesia. Rats in the early training group were implanted with two unilateral arrays of 8 wires aimed at DMS and DLS (0.004' steel wires arranged in a 2 × 4 configuration with wires, each array spaced 2 mm apart, Microprobes). Rats in the extended training group were implanted with two unilateral arrays of 8 electrodes each (0.004' tungsten wires arranged in 2 bundles spaced 2 mm apart and aimed at DMS and DLS) attached to a microdrive allowing the entire array to be lowered by 160 µm increments. For both groups, target coordinates were +0.25 mm AP, +2.3 mm ML, −4.6 mm DV for DMS and +0.25 mm AP, +4.3 mm ML, −4.6 mm DV for DLS. At least 5 days of post-operative recovery commenced before neural recording.

For the inactivation experiments, 26 gauge guide cannulae (Plastic One, Roanoke, Virginia) were implanted bilaterally in 19 rats and targeted to either the DLS (+0.48 mm AP, +4 mm ML, −2 mm DV; final DV: −5 mm) or DMS (+0.36 mm AP, +2.2 mm ML, −3 mm DV; final DV: −4 mm). Inactivation was achieved with a mixture of baclofen (γ-aminobutyric acid-B receptor agonist) and muscimol (γ-aminobutyric acid-A receptor agonist) (B/M: 1.0/0.1 mM; Sigma, St Louis, MO) in a volume of 0.3 µL delivered over 1 min via 33 gauge infusers, 10 min before test. Saline vehicle was administered as

the control condition. All subject were tested in both conditions, with the order of test counterbalanced across subjects.

## Electrophysiological recordings

### Recording

Single-unit activity was acquired during behavior as described previously (*Ottenheimer et al., 2018*; *Richard et al., 2016*; *Richard et al., 2018*), by connecting recording cables to rats' headsets and, at the other end, to a commutator that allowed free movement throughout the recording session. Amplified signals were further processed and stored, along with time stamps of behavioral events, by a multichannel neural recording system (MAP system; Plexon Inc, TX). In the early training group, fixed electrodes allowed recording from the same location throughout training. In the extended training group, the microdrive carrying the electrode arrays was lowered by 160 μm at the end of every other session in order to record from a new set of neurons. In the extended training group, at any electrode location, units were only included from one session for the analysis.

## Analysis of electrophysiological recordings

### Spike sorting

Individual units were isolated offline using principal component analysis in Offline Sorter (Plexon) as described previously (*Ottenheimer et al., 2018*; *Richard et al., 2016*; *Richard et al., 2018*). Auto-correlograms, cross correlograms and the distribution of inter-spike intervals were used to ensure good isolation of single units. Only units with well-defined waveforms and constant characteristics throughout the entire recording session were included in the analyses. Sorted units were exported to Neuro-Explorer (Plexon) for the extraction of neuron and event timestamps, and then onto MAT-LAB (MathWorks) for further analysis.

### Waveform analysis

To dissociate putative-FSI from putative-TAN and MSN, neurons with a firing rate >20 Hz and narrow half-valley width (<0.15 ms) were defined as putative-FSI. Neurons with intermediate characteristics (firing rate between 12.5 and 20 Hz) were unclassified and excluded from analysis. The remaining neurons were defined as putative-TAN/MSN. Given the low number of TAN in dorsal striatum (about 1%) (*Oorschot, 2013*; *Schmitzer-Torbert and Redish, 2008*), we reasoned that including these neurons in our analysis would not significantly impact the pattern of results presented in this study.

### Characterization of task-responsive neurons and z-scores

We assessed neuron responses to each event by conducting t-tests on firing rates during 250 ms periods pre- and post-event, from lever insertion to port entry, in comparison to a 5 s baseline period occurring 20 s prior to each event during the inter-trial interval. Neurons with low baseline firing rate (<0.5 Hz) were not considered. Neurons expressing a significant response (p<0.01) to at least one of the 12 events of the behavioral sequence were considered as task responsive neurons (TRN). Neurons not classified as task-responsive were deemed 'non task-responsive' (nonTRN). Importantly, both TRN and nonTRN were included in classification, decoding and correlation analyses.

The firing rate of each neuron was normalized as follows: $(F_i-F_{mean})/F_{sd}$, where $F_{mean}$ and $F_{sd}$ are the mean and standard deviation of the firing rate during the 5 s baseline period, and $F_i$ is the firing rate at the $i^{th}$ bin of the PSTH. Heatmaps and average PSTHs presented in this study represent the z-scores from −250 ms to 250 ms around each event of the behavioral sequence.

Among TRNs, neurons were considered as excited during the sequence if the average of z-scores from the lever insertion to the port entry was above zero, and were considered as inhibited during the sequence if this average was below zero.

### Classification of distinct neural activity patterns

Phasic and Non-phasic neurons were separated using a Fourier analysis followed by a hierarchical clustering method and the application of a model selection metric (*Figure 3—figure supplement 1*). Specifically, we first reasoned that Non-phasic neurons, expressing by definition sustained

modulation of activity along the behavioral sequence, would be characterized by higher power in low frequency domains (<1 Hz) in comparison with Phasic neurons expressing transient peaks in activity. Note the application of a Fourier analysis here is not to characterize inherent oscillatory activity, but, instead, is a tool to search for distinctions among the time courses of event-evoked spiking patterns. To capture this quantitatively, we computed the power in low (<1 Hz) and intermediate (1–4 Hz) frequency domains for each neuron (fft function in MATLAB; *Figure 3—figure supplement 1*) (for each neuron vector, normalized activity was aligned to each event of the sequence, and the vector represented successive time windows that covered all 7 events of the sequence). We then used these as the features for clustering and applied hierarchical clustering to the dataset (using the linkage function on MATLAB, with the 'ward' parameter; *Figure 3—figure supplement 1C*). Following this, we determined the optimal number of clusters in the dataset by applying the Calinski Harabasz model selection criterion, which assessed the statistical separability of the individual classes obtained for each of 1, 2, 3, ... 10 best clusters identified in the data (CH index; evalclusters function in MATLAB with parameters: 'linkage','CalinskiHarabasz', 'Klist',[1:10]). The optimal number of clusters identified by the CH model selection criterion was 2. The separability of these two clusters was verified using a permutation test (*Figure 3—figure supplement 1F*). We note that our selection of 1 Hz and 4 Hz as low and intermediate frequency limits for extracting features for hierarchical clustering was informed by our original reasoning, and by visual inspection of the data. However, those specific frequency cut-off values were ultimately arbitrary. To explore the sensitivity of our results to the specific values of these cut-offs, we systematically varied both these values (*Figure 3—figure supplement 1E*) and found no substantial impact on the results of clustering: similar outcomes in terms of optimal number of clusters and their separability were obtained in 60% of the cases (*Figure 3—figure supplement 1E–F*).

Following separation of Phasic and Non-phasic neurons, we further classified the Phasic and Non-phasic populations using hierarchical clustering. To account for the strong disparity in z-scores among the population of neurons, activity was normalized as follow: $(F_i - F_{mean})/F_{|max|}$ where and $F_i$ is the firing rate at the $i^{th}$ bin of the PSTH, $F_{mean}$ is the mean of the firing rate during the 5 s baseline period, and $F_{|max|}$ is the absolute maximum neural response. Three features were used in the hierarchical clustering algorithm: the mean normalized activity at the initiation (0–250 ms from lever insertion), execution (250 ms periods around each lever press) and termination (250 ms prior port entry) of the sequence. The number of clusters retained was determined using the Calinski Harabasz criterion.

To investigate the representation of each phasic and non-phasic class during early training, we first separated Phasic and Non-phasic neurons using a Fourier analysis, as described above, and trained a random forest classifier (*Breiman, 2001*; *Liaw and Wiener, 2002*) on the extended training dataset using Phasic (Start, Stop, Middle) or Non-phasic classes (EXC, INH) as labels and the mean normalized activity at the start, middle and end of the sequence, as features. The random forest classifier was then tested on the early training dataset. The 'out of bag' error estimates from the Random Forest classifier were 0.036 and 0 for Phasic and Non-phasic classes, respectively, which constitute low values for generalization error providing an additional validation of the classification approach used here.

## Decoding

A linear discriminant analysis (LDA) model (the 'fitcdiscr' function in MATLAB) was trained on mean peri-event z-scores (from lever insertion to port entry; vector of 12 elements) for 90% of the full dataset (i.e., putative MSNs). This model was then used to classify the remaining 10% of individual neurons as belonging to DLS or DMS. This was repeated for each 10% of the dataset in a cross-validation approach. To account for the unbalanced number of recorded neurons in DLS versus DMS, we performed the analysis on the same number of neurons in each region by randomly sampling neurons in DMS based on the number of recorded DLS neurons. We performed this analysis on 50 random selections of neurons pooled from the different rats on a given session, and averaged performance across all 50 repetitions to determine model accuracy. We also conducted the same analysis with the region identities shuffled to determine accuracy expected by chance. We compared the mean decoding accuracy across early training sessions. During extended training, we performed the analysis on 50 random selections of different ensemble sizes of neurons (60, 120, 200, 400, 600

neurons), pooled from different sessions and rats. Decoding accuracy was compared across ensemble sizes.

To account for subtle modulations of activity along the behavioral sequence not captured by mean peri-event z-scores, we conducted a principal component analysis (PCA) on the concatenated z-scores around each event of the behavioral sequence (−250 to 250 ms, 10 ms bins, seven events *51 bins vector) and trained LDA models on these principal components for each early training session and during extended training. We first compared the decoding accuracy as a function of the number of principal components included in the analysis. This allowed us to determine the minimal number of principal components necessary to reliably decode brain regions in both dataset, as a function of sessions or ensemble sizes (*Figure 7—figure supplement 1*). We then repeated this analysis using neurons' first three principal components, for each early training session and during extended training.

To compare accuracy in region decoding across sessions and ensemble size, we performed permutation tests for shuffled and true data, in each early training session and for each ensemble size during extended training.

### Correlation analysis and shuffled data controls

To assess relationships between neural activity of individual neurons and behavior, we conducted Spearman's rank correlations. We examined the trial-by-trial correlation between firing rate at the initiation (0–500ms post lever insertion), execution (from $1^{st}$ to last lever press) and termination (0–500ms post lever retraction) of the sequence with the first lever press latency, response rate, and port-entry latency, respectively. Neural activity during execution of the lever press sequence was computed by normalizing the number of spikes between the first and last lever press to the sequence duration. As in *Richard et al. (2018)*, to control for significant correlations occurring spuriously in a small percentage of units, we ran Spearman correlations on 1000 iterations of shuffled data for each behavioral variable (*Richard et al., 2016*; *Richard et al., 2018*). The trial-by-trial latencies or response rate were randomly shuffled for each neuron for each iteration, and Spearman correlations that related behavior and firing rate during the relevant event window were assessed for each neuron. We then evaluated the distribution of both the number of units with significant correlations (at p<0.05) and the mean correlation coefficient to determine to what extent these variables stand apart from the shuffled data distribution (*Figure 10—figure supplements 2–4*).

### Statistical analysis

Mean response rate (in resp/s) was measured as the number of lever presses per trial, divided by the time of lever availability and averaged across trials. Data following a normal distribution were subjected to repeated measures analysis of variance (ANOVA). Appropriate non-parametric tests (sign-test, Kolmogorov-smirnov) were used when normality assumption was violated. For neural population measures, mean z-scores were computed by averaging time bins from 0 to 250 ms pre and post-events from lever insertion to the port entry following reward delivery. Mean normalized population activity in DMS and DLS was compared across events using repeated-measures ANOVA (with Geisser-Greenhouse correction for violation of sphericity), with events as the within factor and region as the between factor. During early training, sessions were considered as a between factor. Proportions of neurons were compared using chi-squared tests. Statistical analyses were performed on MATLAB (MathWorks) and Statistica (StatSoft 7.0).

## Histology

To verify the electrodes and cannulas placement, animals were deeply anesthetized with pentobarbital and, when appropriate, electrode sites were labeled by passing a DC current through each electrode. All rats were perfused intracardially with 0.9% saline followed by 4% paraformaldehyde (extended training group with tungsten wires), or 4% paraformaldehyde with 3% potassium ferricyanide (early training group with steel wires). Brains were removed, post-fixed in 4% paraformaldehyde for 4–24 hr, cryo-protected in 20% sucrose for >48 hr, sectioned at 50 μm on a cryostat, and stained with cresyl violet.

## Additional information

### Funding

| Funder | Grant reference number | Author |
|---|---|---|
| National Institute for Health Research | R01DA035943 | Patricia H Janak |
| National Institute on Alcohol Abuse and Alcoholism | R01AA026306 | Patricia H Janak |

The funders had no role in study design, data collection and interpretation, or the decision to submit the work for publication.

### Author contributions

Youna Vandaele, Conceptualization, Data curation, Formal analysis, Investigation, Methodology, Writing—original draft, Writing—review and editing; Nagaraj R Mahajan, David J Ottenheimer, Formal analysis, Methodology, Writing—review and editing; Jocelyn M Richard, Methodology, Writing—review and editing; Shreesh P Mysore, Conceptualization, Resources, Supervision, Methodology, Writing—review and editing; Patricia H Janak, Conceptualization, Resources, Supervision, Funding acquisition, Investigation, Methodology, Writing—original draft, Writing—review and editing

### Author ORCIDs

Youna Vandaele (iD) https://orcid.org/0000-0002-8389-8850
Nagaraj R Mahajan (iD) https://orcid.org/0000-0002-4437-2645
David J Ottenheimer (iD) https://orcid.org/0000-0003-4882-1898
Jocelyn M Richard (iD) https://orcid.org/0000-0001-5750-0418
Shreesh P Mysore (iD) http://orcid.org/0000-0002-7781-8252
Patricia H Janak (iD) https://orcid.org/0000-0002-3333-9049

### Ethics

Animal experimentation: This study was carried out in accordance with the recommendations of the Guide for the Care and Use of Laboratory Animals (National Research Council, 1996), and was approved by the institutional animal care and use committee of Johns Hopkins University.

### Decision letter and Author response

Decision letter https://doi.org/10.7554/eLife.49536.027
Author response https://doi.org/10.7554/eLife.49536.028

## Additional files

### Supplementary files

• Transparent reporting form DOI: https://doi.org/10.7554/eLife.49536.023

### Data availability

Behavioral and single unit recording data have been deposited on G-Node, as well as the MATLAB codes used to analyze the data and generate the figures.

The following dataset was generated:

| Author(s) | Year | Dataset title | Dataset URL | Database and Identifier |
|---|---|---|---|---|
| Vandaele Y, Mahajan NR, Ottenheimer DJ, Richard JM, Mysore SP, Janak PH | 2019 | Distinct recruitment of dorsomedial and dorsolateral striatum in an instrumental task erodes with extended training | http://doi.org/10.12751/g-node.92538d | G-Node, 10.12751/g-node.92538d |

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
