## [Decision Letter]

Thank you for submitting your article "Distinct recruitment of dorsomedial and dorsolateral striatum erodes with extended training" for consideration by *eLife*. Your article has been reviewed by three peer reviewers, including Naoshige Uchida as the Reviewing Editor and Reviewer #1, and the evaluation has been overseen by Kate Wassum as the Senior Editor. The following individual involved in review of your submission has agreed to reveal their identity: David Robbe (Reviewer #2).

The reviewers have discussed the reviews with one another and the Reviewing Editor has drafted this decision to help you prepare a revised submission.

Summary:

Vandaele and colleagues monitored the activity of neurons in the dorsomedial and dorsolateral striatum (DMS and DLS, respectively) while rats were trained in a discrete trials fixed ratio-5 (DT5) procedure. A prevailing view in the field is that DMS and DLS are involved in goal-directed and habitual behaviors, respectively, and training shifts the involvement of DMS to DLS as the behavior becomes more habitual. Furthermore, patterns of activity related to sequence learning such as the appearance of start and end of motor sequence and sustained activity between them are thought to underlie habitual motor behaviors. The present study found that although neuronal activity in DMS and DLS differ early during training, their activity patterns become similar after overtraining. The data also showed that the activity of some neurons in both DMS and DLS was correlated with movement parameters, and inactivation of either DMS or DLS affected the animal's performance, indicating that these regions are involved in the control of behaviors.

These results challenge the prevailing idea regarding the activity patterns related to habitual behavioral control as well as the involvement of these areas in habit learning. The reviewers pointed out potential weaknesses such as that (1) the behavior of the animal becomes insensitive to the devaluation test (i.e. habitual) relatively early and it is unclear whether the authors were able to record in the period during which the behavior is clearly goal-directed, and therefore, that (2) the authors were unable to identify particular features of neuronal activity that correlated with transition from goal-directed to habitual behaviors. Conversely, the authors were unable to identify what features of behaviors are correlated with the changes in neuronal activity during overtraining.

The reviewers' overall evaluations varied somewhat, with some reviewers being more critical of the caveats mentioned above. However, all the reviewers thought that the results, in particular, the convergence of activity patterns between DMS and DLS after overtraining, are surprising and agreed that this study contains important results that will challenge the prevailing views regarding how DMS and DLS regulate goal-directed and habitual behaviors and subtleties of prevailing paradigms and interpretations of the previous results.

Essential revisions:

1) The authors discuss "engagement" of DMS and DLS in habit formation. For instance, in the Abstract, the authors state that "These results suggest that behavioral sequences may continue to engage both striatal regions long after initial acquisition, when performance is highly regular and habitual". However, later in the manuscript, the authors also point out the difficulty of relating the patterns of activity observed in the data and specific phases of goal-directed to habitual transition. In the last paragraph of Discussion, the authors discuss this issue clearly: "Thus, it is not clear whether or how sequence-related activity relates to either skill learning/performance or habit learning/performance". The reviewers found this apparent discrepancy very confusing. First, are there any behavioral changes that occur during overtraining? The reviewers thought that other behavioral measures might be better for evaluating habit or the transition from the early to over-trained phases. Second, the reviewers found that the last sentence of Discussion is insightful and provides a good summary of the implication of the study. We suggest that the authors emphasize the content discussed in this paragraph, and rewrite the Results and other parts of Discussion sections so that they are more consistent with one another.

2) The reviewers thought that additional experiments can clarify some of the remaining issues. In particular, showing recording data from probe tests and trials when the sequence was not completed, showing recording data from even earlier in training (e.g. DT1), looking at side-by-side-recordings from rats doing free-running FR5, etc. (Please see the individual comments for detail). Although these experiments are not required in light of *eLife*'s policy for requested revisions to be feasible within an approximate 2 month time frame, we urge the authors to consider these experiments, or at least discuss in the manuscript.

3) The reviewers pointed out various issues in the data analysis (see below the individual comments). We request that the authors address these issues by additional analyses and revising the manuscript.

4) It is unclear at what point in training the DMS and DLS inactivations were done (i.e. after how many weeks/sessions of DT5 training). Please clarify. If it was after very extended DT5 training then this is really an important point: different than previous lesion studies, and important for their argument that DMS and DLS are still engaged in the task after the long extended training.

5) The analysis comparing the activity patterns between goal-directed and habitual animals is very informative. We request that this analysis (Supplementary Figure 12) be moved to a main figure.

Overall, we think that the authors can address the essential points and support the authors' main conclusions without additional experiments. Additional experiments are not required but would be very useful in clarifying specific issues.

Reviewer #1:

Previous studies have indicated that the dorsomedial striatum (DMS) is involved in goal-directed behaviors while dorsolateral striatum (DMS) is involved in habitual behaviors. A common view is that behavioral overtraining shifts the involvement of DMS to DLS. In accord with this, several studies have indicated that patterns of neural activity that are thought to bracket or chunk a sequence of movements appear in the DLS after overtraining. Some recent studies, however, have provided data indicating that a simple dichotomy may not describe the functions of DMS and DLS accurately. It is therefore important to clarify whether and in what conditions these ideas hold.

In this study, Vandaele and colleagues addressed this question by recording the neuronal activity in the DMS and DLS while rats performed a discrete trials fixed ratio (DT5) task. In this task, rats have to press a lever five times before obtaining a food reward. A lever insertion signaled rats to initiate lever pressing and the lever was retracted immediately after a fifth-lever press. This task design builds on their previous work that showed that the insertion/retraction promotes habit formation compared to a similar task with a free-standing lever. In the present data set, the lever pressing behavior became more rapid and stereotyped over the course of ~10 days of training. After extended training (8 weeks), the lever pressing behavior became insensitive to devaluation, indicative of habit formation.

During initial training (<10 days of training), the authors observed a big difference in neuronal activity between the DMS and DLS. Many DMS neurons exhibited sustained inhibition while DLS neurons showed sustained excitation. Transient excitations at the initiation and termination of lever presses and sequential activations during task performance were observed in both areas. These differences were evident at the gross population level, at the individual neuron level, and based on a decoding analysis. Surprisingly, these differences diminished after extended training. Finally, inactivation of DMS and DLS impaired the stereotyped performance in a similar manner. These results suggested that habit formation and the appearance of task-bracketing activities can be dissociated depending on task conditions. The authors discuss potential reasons why the present results differ from previous studies, such as the role of cue (lever insertion/retraction) in triggering phasic responses in DMS.

Overall, the results are surprising and provides important insights into how DMS and DLS regulate operant behaviors. The inclusion of the data from extended training is illuminating. The data analyses were done at different levels in a compelling way and they together make a compelling case to support the authors' conclusion. The exact reason why the results are different from previous studies remains unclear but the authors make interesting discussions. This study raise various important issues that will open up future investigations.

I have some relatively minor technical concerns which are unlikely to change the authors' main conclusions but are important for understanding the data.

1) The exact reason why the present result differed from the previous work remains unclear. Another possibility is that the animal in this task does not really have to "know" how many lever presses are required to get reward because the lever is retracted immediately after the 5th lever press. It would be illuminating to include probe trials in which lever retraction is delayed. Does the animal leave the lever without retraction or does the leaving depend on retraction? The authors provide evidence supporting habitual nature of behavior (stereotypy, speed, and devaluation) but it would be interesting to discuss this point.

2) The authors use a Fourier analysis to first classify neurons into phasic versus non-phasic types. It remains somewhat unclear what aspects of neuronal activity this analysis actually captures. It would be useful to analyze and report this issue more carefully. (1) the non-phasic type or "sustained" activity may actually consist of multiple phasic activities. Are there neurons that exhibit multiple phasic activities (e.g. every lever press) in the data set? If so, how are these neurons classified? The analysis was done by averaging across trials, and there is a possibility that it may smear out phasic responses. Clarifying these issues would facilitate better understanding of the analysis. (2) In the panel showing the firing patterns of "phasic neurons" (Figure 3—figure supplement 1 and Figure 6—figure supplement 1), the neurons at the bottom (and the top) appear to have sustained activity although these neurons may be exhibiting multiple phasic responses. Please clarify. (3) The authors included non-responsive neurons in this analysis. Some of the "inhibited" or "excited" neurons may have actually been not responsive. (4) Some neurons appear to show both phasic and sustained responses. I understand that the number of clusters are verified by the Calinski Harabasz criterion. But this does not necessarily exclude the presence of these neurons.

3) Throughout the manuscript, the authors indicate that their analyses are "unbiased". It is unclear in what sense they are unbiased. The authors are making a number of choices toward particular analysis or criterion. Perhaps, use "objective"? or clarify in what sense unbiased.

4) In the correlation between neuronal activity and three phases of behavior (1st lever press latency, response rate, and port entry latency), the authors define the "execution" window as the window between the first and the last lever press. This is a variable window and therefore, will be highly anti-correlated with response rate even where there is no firing modulation. This analysis has to be done and reported carefully.

5) It is unclear from the results "monitoring" is a good way to characterize the activity correlated with behaviors. It is possible that these activities are controlling behaviors. Why does the author use "monitoring"? That being said, a similar idea was proposed by Rueda-Orozco and Robbe, 2015. It would be good to cite this paper.

Reviewer #2:

In this manuscript, Vandaele et al., recorded spiking activity simultaneously in the dorsomedial and dorsolateral regions of the striatum (DMS, DLS) during learning and performance of lever press sequence task. The authors wanted to examine the validity of a classical theory in the field: the DMS contributes to early phases of task learning while DLS contribute to performance once the task is habitual. The firing pattern of DMS and DLS population was distinct but did not evolve much during learning although they were more similar when recordings were performed after extensive training. The authors concluded that contrary to the classical view both regions are engaged throughout learning and long after learning.

The general conclusion is interesting and could be novel enough to deserve publication in *eLife*. But in my opinion there many contradictory results and odds/unclear analyses. In particular, the authors developed a functional classification of the neurons (phasic, no-phasic; start, stop, middle) which is statistically contestable and is used extensively in the manuscript. More important (and again in my opinion) this analysis does not bring much more information or additional evidence in regard to the conclusion of the paper, as mentioned in the Abstract. I also find that some important observation which contradicts previous classical studies are not sufficiently emphasized. Finally, I have a strong technical concern I would like the author to clarify.

1) The authors claim that they use an unbiased classification of their neurons but in fact, many aspects of this classification are arbitrary and have unclear statistical ground. The authors decided to use the power in low (<1Hz) and "intermediate" (1-4Hz) frequency domains of each neuron spike train as features for hierarchical clustering. To the best of my knowledge, it is the first time such a method is used (at least the authors do not provide references). First, this choice of frequency is arbitrary. Second power in spike train is quite a tricky measure and is strongly sensitive to the total numbers of spikes recorded. |I doubt this is a good method for low firing rates observed in the striatum. In support of my doubt, the average power spectrum shown in Figure 3—figure supplement 1 peak at the lowest frequency value (first bin after 0 Hz). I don't think it is statistically valid to use power spectrum analysis while a majority of neurons do not show clear oscillatory patterns. In other words, the clustering algorithm is applied to noisy data with arbitrary threshold. Third, it is not because a clustering algorithm returns an optimal number of clusters that it means that data can be statistically separated. This requires combining the clustering algorithm with a permutation of the data points (shuffling of the paired value) and bootstrapping. Without showing such statistical disambiguation the authors cannot reject the hypothesis that there is no discrete categories but rather a continuum of firing rate profiles. By looking at the data in Figure 3—figure supplement 1 it appears that there is significant overlap between the 2 main categories: the phasic neurons in the upper part of Figure 3—figure supplement 1D are more similar to non-phasic neurons in the upper panel of Figure 3—figure supplement 1E, than to phasic neurons in the lower part of Figure 3—figure supplement 1D.

Without demonstrating that there are discrete functional categories of neurons (which is very unlikely by looking at the data in Figure 2), the authors should rewrite a large section of their manuscript and consider the possibility that striatum encodes continuously sensorimotor dynamics. This is an important point as this issue has been raised recently (a paper cited by the authors Sales-Carbonell et al., 2018 but see also Robbe et al., 2018). Also, instead of using oscillatory analysis as features characterizing neuronal activity, why not using the averaged and normalized perivent histograms (data in Figure 2D, E) and compute PCA on these to characterize two or three values that capture most of the variability of the peri-event histograms (see Sales-Carbonnel, 2018).

2) In the Abstract, the authors stated that DMS and DLS are "engaged" during learning. This statement is a bit contradictory with the fact that firing patterns in both regions do not evolve while animals improve in the task. Again, this is especially contradictory as the authors emphasized the existence of star and stop neurons. Usually, a causal role in behavior is inferred when there is a change in neuronal activity that correlates with a change in task proficiency (e.g., Barnes et al., 2005, Jin and Costa, 2015).

3) The results of the inactivation experiments do not fit very well with the general spin of the paper as there is no strong effect on the number of lever press. The strongest effect seems to be an increase in the latency to 1st press and a decrease in response rate. Again, this is difficult to reconcile with striatal neurons starting or maintaining or stopping sequences. On the other hand, this result with work emphasizing a role of the striatum in vigor/movement speed (Kim et al., 2014, Rueda-Orozco et al., 2015, Panigrahi et al., 2015).

4) One of the most surprising aspects of this data is that the population of DLS neurons fire continuously during the sequence. This is in plain contradiction with the start and stop hypothesis or the task bracketing hypothesis (especially it contradicts Martiros et al., 2018 paper that used a very similar lever press task). On the other hand, this result is very similar to Sales-Carbonnel et al., 2018 and Rueda-Orozco and Robbe 2015). I am really surprised that the authors do not emphasize such an important result.

5) The title of the paper is "Distinct recruitment of dorsomedial and dorsolateral striatum erodes with extended training". And in the Abstract it is stated that DMS and DLS activity converged after extensive. First, the authors have no direct evidence for such convergence as they did not record from naive to extensive. During the first 10 sessions (at the end of which performance plateaued) there is no sign of convergence. Second, it seems from Figure 2C that the increased similarity in coding arise from diminished modulation of both DLS and DMS population activity (Z score closer to zero). I am not sure that the authors can conclude in some kind of functional convergence while both regions are becoming less modulated.

6) The authors stated that they recorded in average 81 units per session in the DMS with 8 single electrodes implanted. That means they recorded routinely 10 well-isolated units per electrode. I doubt this is possible. Could the authors provide a view of these well-isolated clusters showing how they are separated from noise/MUA?

Reviewer #3:

The authors show that, although previous literature has shown the dorsomedial striatum (DMS) mediates goal-directed actions and the dorsolateral striatum (DLS) mediates habitual behaviors, the DMS remains engaged even after habit development. The recordings were done using a relatively novel behavioral task, which the researchers published on in 2017. The data are intriguing and represent an important challenge to current theories of habit formation, yet they are purely correlational. However, the lack of causal data might be balanced by a more thorough analysis of the recording data in correlation with specific behavioral events, and in this way the work could still be quite impactful as it challenges current theories of habit formation. We provide some suggestions for improvement below:

• The authors present data from an early training and extended training group of rats, yet there is not a clear justification of the time points used in these groups and no discussion of how the reader should interpret these different timepoints in relation to our broader understanding of habit. If behavior stabilizes by the end of early training, yet neural activity changes by the time the extended training group is recorded, how are we to interpret these results? Is it possible the data recorded in the early training group are correlated with habit acquisition while the data recorded in the extended training group are due to later consolidation? A discussion on what the change in signal represents (presumably not changes in behavior) would benefit the overall interpretation of the observed changes in DMS and DLS signals.

• The lack of data from a time point where the animals are not habitual limits the interpretations that can be made about signal changes that occur as habit forms.

• Figure 1C serves as the single piece of data to show animals are developing habit – however the authors note that the difference between valued and devalued responding is p=.06 which could be considered marginally significant. Looking at the individual data points reveals several animals that display noticeable decreases in lever pressing under the devalued condition, suggesting these animals are not yet habitual. Likewise, in the extended training group there are also multiple animals with noticeable decreases. The authors do make some attempt at analysis of the individual data in Supplementary Figure 12. We would suggest that this be moved into the main figures and analyzed further, as the results in the goal-directed vs. habitual rats are not identical. If the satiation probe is not reliably distinguishing differences, but is only good for distinguishing group averages (e.g. perhaps because the individual differences are simply due to an order effect of testing that needs to be counterbalanced in the group?), this limits the analyses that can be done, and should be noted.

• Some rats are trained with sucrose and some with pellets. The authors previously showed that reward type mattered for habit formation (Vandaele et al., 2017). It is not clear in this manuscript which rats were trained using which reward and whether that affected any of the results.

• Were recordings done during the probe sessions? These would be interesting to analyze and present, especially with respect to the individual differences in performance on the probe trials.

• Although clearly somewhat rare, it would be interesting to see what happens in these recordings on trials in which the rats failed to complete the 5 presses (failure to complete ratio with 1 minute).

• We wonder whether other outcome measures might be better for evaluating habit. It would be informative to see response rates and CV within-sequence for the probe trials. It would also be good to show 1st lever press latency in Figure 1 for comparison with Figure 8.

• In Figure 8 the authors use 1st lever press latency as a measure of habit, however this measure is not presented as a reliable measure of habit previously in the manuscript. Also in Figure 8C-F – why are the same outcome measures not shown for both DMS and DLS?

• The characterization of neurons used throughout the manuscript lacks nuance in terms of what these neurons may be doing. For instance, in Figure 3 the authors categorize neurons as either phasic start or stop neurons depending on their peak activity during the task, however the stop neurons look like they may also be signaling the port entry – a more careful timing analysis of this data like an analysis of recordings during a port entry that doesn't follow sequenced nose poking would help explain the nature of this late increase. Additionally, several neurons within the heatmaps seem to fall under multiple categories (start and middle; middle and stop; start, middle, and stop) – however there is no mention of neurons that are active across multiple behavioral time points. In Figure 5, it is noticeable that similarly categorized DMS and DLS neurons do look very different from each other in terms of overall activity.

• Interpretation of the results is limited by the fact that direct and indirect pathway neurons cannot be distinguished. Also, unless we are missing something, it does not appear that putative FSIs and unidentified neurons are included in the analyzed data.

Overall, this work has the potential to be very informative for understanding the role of striatal subregions in habitual responding. However, although the authors mention these findings are at odds with previous literature, they stop short of explaining why this may be or how we should view this circuit going forward in light of their findings, which ultimately limits the impact and significance of their work.

---

## [Author Response]

Essential revisions:1) The authors discuss "engagement" of DMS and DLS in habit formation. For instance, in the Abstract, the authors state that "These results suggest that behavioral sequences may continue to engage both striatal regions long after initial acquisition, when performance is highly regular and habitual". However, later in the manuscript, the authors also point out the difficulty of relating the patterns of activity observed in the data and specific phases of goal-directed to habitual transition. In the last paragraph of Discussion, the authors discuss this issue clearly: "Thus, it is not clear whether or how sequence-related activity relates to either skill learning/performance or habit learning/performance". The reviewers found this apparent discrepancy very confusing. First, are there any behavioral changes that occur during overtraining? The reviewers thought that other behavioral measures might be better for evaluating habit or the transition from the early to over-trained phases.

We did not find any obvious behavioral changes occurring during overtraining. We observed that some measures such as the response rate, the coefficient of variation of response rate, and the first lever press latency, reached an asymptotic level after extended training (Figure 1E-G). The first lever press latency across early training sessions and after extended training is now illustrated in Figure 1G on the recommendation of reviewer 3. But in answer to the broader question, it is not clear from this data set how the observed neural activity patterns relate to improved performance/more automatic performance over time. Because the activity patterns are by definition correlated in time with the occurrence of the salient events/actions including lever extension, lever presses, lever retraction and port entry, such neural activity changes are typically interpreted as critical for behavior production, behavior monitoring, or behavior feedback, and while the trial-by-trial correlations between neural firing and behavioral measures support this connection with real-time behavior, we can’t demonstrate a clear connection with the nature of the control of that real-time behavior, i.e., habitual or not, through observation of the activity patterns. Although there is not a qualitative change in the behavioral measures from the end of initial training to extended training, the changes in neural activity patterns over this period of time could, as a reviewer suggested, reflect changes in representation as consolidation processes develop over time. We are now better describing and discussing these important findings throughout the manuscript.

Second, the reviewers found that the last sentence of Discussion is insightful and provides a good summary of the implication of the study. We suggest that the authors emphasize the content discussed in this paragraph, and rewrite the Results and other parts of Discussion sections so that they are more consistent with one another.

We are now emphasizing the content of the last paragraph of the Discussion throughout the manuscript. In line with the first comment of reviewer 3, we emphasized the improvement in performance across early training sessions and suggest that skill consolidation is occurring after extended training. We are putting less emphasis on relations between neural activity and habitual/skill learning and rather suggest correlations with performance. We underline in the Results and Discussion the difficulty in relating activity in dorsal striatum with habitual learning in these subjects.

2) The reviewers thought that additional experiments can clarify some of the remaining issues. In particular, showing recording data from probe tests and trials when the sequence was not completed, showing recording data from even earlier in training (e.g. DT1), looking at side-by-side-recordings from rats doing free-running FR5, etc. (Please see the individual comments for detail). Although these experiments are not required in light of eLife's policy for requested revisions to be feasible within an approximate 2 month time frame, we urge the authors to consider these experiments, or at least discuss in the manuscript.

We are now introducing data recorded during DT1 early training sessions in Figure 2—figure supplement 2, which reveals an early change in DLS activity before the first lever press. We also discuss the need for additional experiments aimed at comparing DT5 recording results with neural activity during training in the free-running FR5 task.

3) The reviewers pointed out various issues in the data analysis (see below the individual comments). We request that the authors address these issues by additional analyses and revising the manuscript.

The different issues in data analysis have been addressed in various parts of the manuscripts. See below for specific details.

4) It is unclear at what point in training the DMS and DLS inactivations were done (i.e. after how many weeks/sessions of DT5 training). Please clarify. If it was after very extended DT5 training then this is really an important point: different than previous lesion studies, and important for their argument that DMS and DLS are still engaged in the task after the long extended training.

This information is now directly specified in the Results subsection “Pharmacological inactivation of both DLS and DMS interferes with task performance”. The DMS group received 20 days of DT5 training and the DLS group received 10 DT5 sessions. These groups cannot be considered as having received an extended training. This experiment is now presented in the aforementioned subsection to prevent any confusion.

5) The analysis comparing the activity patterns between goal-directed and habitual animals is very informative. We request that this analysis (Supplementary Figure 12) be moved to a main figure.

As requested, Supplementary Figure 12 in our original manuscript has been moved to the main text as Figure 8 and the results of the statistical analysis are now detailed in the subsection “Individual differences in sensitivity to outcome devaluation do not substantially correlate with dorsostriatal activity”.

Overall, we think that the authors can address the essential points and support the authors' main conclusions without additional experiments. Additional experiments are not required but would be very useful in clarifying specific issues.

We thank the reviewers and editors for the evaluation. We have addressed each comment to the best of our ability, and discuss our preference regarding additional experiments.

Reviewer #1:Previous studies have indicated that the dorsomedial striatum (DMS) is involved in goal-directed behaviors while dorsolateral striatum (DMS) is involved in habitual behaviors. A common view is that behavioral overtraining shifts the involvement of DMS to DLS. In accord with this, several studies have indicated that patterns of neural activity that are thought to bracket or chunk a sequence of movements appear in the DLS after overtraining. Some recent studies, however, have provided data indicating that a simple dichotomy may not describe the functions of DMS and DLS accurately. It is therefore important to clarify whether and in what conditions these ideas hold.In this study, Vandaele and colleagues addressed this question by recording the neuronal activity in the DMS and DLS while rats performed a discrete trials fixed ratio (DT5) task. In this task, rats have to press a lever five times before obtaining a food reward. A lever insertion signaled rats to initiate lever pressing and the lever was retracted immediately after a fifth-lever press. This task design builds on their previous work that showed that the insertion/retraction promotes habit formation compared to a similar task with a free-standing lever. In the present data set, the lever pressing behavior became more rapid and stereotyped over the course of ~10 days of training. After extended training (8 weeks), the lever pressing behavior became insensitive to devaluation, indicative of habit formation.During initial training (<10 days of training), the authors observed a big difference in neuronal activity between the DMS and DLS. Many DMS neurons exhibited sustained inhibition while DLS neurons showed sustained excitation. Transient excitations at the initiation and termination of lever presses and sequential activations during task performance were observed in both areas. These differences were evident at the gross population level, at the individual neuron level, and based on a decoding analysis. Surprisingly, these differences diminished after extended training. Finally, inactivation of DMS and DLS impaired the stereotyped performance in a similar manner. These results suggested that habit formation and the appearance of task-bracketing activities can be dissociated depending on task conditions. The authors discuss potential reasons why the present results differ from previous studies, such as the role of cue (lever insertion/retraction) in triggering phasic responses in DMS.Overall, the results are surprising and provides important insights into how DMS and DLS regulate operant behaviors. The inclusion of the data from extended training is illuminating. The data analyses were done at different levels in a compelling way and they together make a compelling case to support the authors' conclusion. The exact reason why the results are different from previous studies remains unclear but the authors make interesting discussions. This study raise various important issues that will open up future investigations.I have some relatively minor technical concerns which are unlikely to change the authors' main conclusions but are important for understanding the data.1) The exact reason why the present result differed from the previous work remains unclear. Another possibility is that the animal in this task do not really have to "know" how many lever presses are required to get reward because the lever is retracted immediately after the 5th lever press. It would be illuminating to include probe trials in which lever retraction is delayed. Does the animal leave the lever without retraction or does the leaving depend on retraction? The authors provide evidence supporting habitual nature of behavior (stereotypy, speed, and devaluation) but it would be interesting to discuss this point.

Absolutely. Our interpretation of behavioral findings in the DT5 task is that rats rapidly develop habit and behavioral chunking specifically because they are not required to track how many times to press the lever to get the reward. Rats can merely continue pressing until the retraction of the lever. We predict that during probe trials in which lever retraction is delayed, rats would keep pressing on the lever until its retraction. Indeed, using a task similar to the DT5 procedure, in which reward delivery was signaled by the lever retraction, we found that rats made on average 3.3 ± 0.5 additional lever presses during such probe trials, where lever retraction was delayed by 2 second from the fifth lever press.

We are now discussing this point in the Discussion: “In fact, in the DT5 task, rats do not have to track the number of presses emitted to obtain the reward and may continue pressing on the lever until its retraction, which may favor behavioral chunking and the specific activity pattern reported in this study”.

2) The authors use a Fourier analysis to first classify neurons into phasic versus non-phasic types. It remains somewhat unclear what aspects of neuronal activity this analysis actually captures. It would be useful to analyze and report this issue more carefully.a) The non-phasic type or "sustained" activity may actually consist of multiple phasic activities. Are there neurons that exhibit multiple phasic activities (e.g. every lever press) in the data set? If so, how are these neurons classified?

Neurons showing significant responses to all 5 lever presses (both before and after lever presses) were found in both phasic and non-phasic classes: these neurons represented 41% of non-phasic neurons (N=46) and 1.4% of phasic neurons (N=9), when the activity pre- and post-lever press was compared to the baseline activity during inter-trial intervals. However, it is not possible from this analysis to determine whether neurons’ response to lever press events results from phasic or sustained activity. Analysis of the activity of the 9 phasic neurons responding to each lever press reveals that these few neurons appear to express sustained excitation or sustained inhibition during lever pressing, and their classification as phasic resulted from a large peak after the lever insertion and/or after the last lever press. The PSTHs in Author response image 1 illustrate the difference between neurons responding to every lever presses and classified as phasic or non-phasic. As the reviewer can see, Non phasic neurons expressed sustained excitation or sustained inhibition without peaks post lever insertion or post Lever retraction, which may explain why there were classified as Non phasic. This is now specified in the last paragraph of the Results subsection “Classification of distinct neural signatures in the dorsal striatum during extended sequence training”.

The analysis was done by averaging across trials, and there is a possibility that it may smear out phasic responses. Clarifying these issues would facilitate better understanding of the analysis.

We do average across trials, but keep the time relative to the events. To create the vector for each neuron, normalized activity was first aligned to each event of the behavioral sequence, with time windows of -0.25s to 0.25s around events and time bins of 0.01s. The Fourier analysis was then conducted on one vector per neuron that represented the successive time windows that covered all 7 events of the behavioral sequence. Therefore, phasic activity represents neuronal responses to specific events. This is now clarified in the Materials and methods subsection “Classification of distinct neural activity patterns”.

As suggested by the reviewer, high response rates in the DT5 task could in theory confound our classification of phasic and non-phasic neurons; phasic responses to every lever presses could appear as a sustained response. However, if this were true, one would expect the number of neurons classified as non-phasic to be correlated with the response rate. Although we observed a rise in response rate across early training sessions, we did not observe any change in the proportion of non-phasic neurons (Figure 6—figure supplement 1B, χ^2^=9.0, p=0.437). Furthermore, we did not observe clear changes in the activity of neurons showing sustained excitation or sustained inhibition across early training sessions.

To further examine whether sustained activity may result from multiple phasic activations around the lever presses, we reproduced Figure 6 using peri-event time windows of 1.5s (from -0.75 to 0.75s around each events) instead of 0.5s (-0.25 to 0.25s). As can be seen in Author response image 2, phasic excitation at the lever insertion and at the termination of the sequence in phasic start and stop neurons was repeated from one event to the next one, but the activity returned to baseline before the next event. In contrast, the activity in non-phasic neurons, while certainly dynamic, does not return to baseline during the extended time window, even when lever presses were on average 1-2 seconds apart during the first three early training sessions (Day 1-3, response rate between 0.5-1 response/sec). These results suggest that sustained activity in non-phasic neurons does not result from multiple phasic responses around each lever press.

**Author response image 2. respfig2:** 

b) In the panel showing the firing patterns of "phasic neurons" (Figure 3—figure supplement 1 and Figure 6—figure supplement 1), the neurons at the bottom (and the top) appear to have sustained activity although these neurons may be exhibiting multiple phasic responses. Please clarify.

Multiple **phasic** responses to every lever press events occurred rarely (see response to previous comment). However, although some phasic neurons appear to have sustained activity and fire above baseline during inter-response intervals over the 5 lever presses, this population only represented 1.4% of the phasic neurons, and typically included units that also displayed activity at lever insertion or retraction. Therefore, we think the occurrence of strong transient responses prominent at the start of the behavioral sequence (after the lever insertion) and at the end of the sequence (after the last lever press or before the port entry) are responsible for the classification of these neurons as phasic. This results in the presence of neurons classified as phasic but showing both sustained and phasic responses, as highlighted by the reviewer in the following comment. This observation is now included in the last paragraph of the subsection **“**Classification of distinct neural signatures in the dorsal striatum during extended sequence training”.

c) The authors included non-responsive neurons in this analysis. Some of the "inhibited" or "excited" neurons may have actually been not responsive.

The reviewer is correct, we included non-task responsive neurons (as determined by traditional statistical analysis of activity around events vs. baseline activity) in the analysis to remain as objective as possible in our classification approach. We avoided pre-selecting neurons based on other arbitrary criterion. This represents our attempt to analyze the data using multiple approaches that are not dependent on one another. It is however important to point out that non-responsive neurons represented a minority of putative-MSNs in our study.

d) Some neurons appear to show both phasic and sustained responses. I understand that the number of clusters are verified by the Calinski Harabasz criterion. But this does not necessarily exclude the presence of these neurons.

We agree – some neurons, for example, showed sustained inhibition during lever pressing with a phasic excitation at the end of the behavioral sequence. In fact by varying the classification parameters (lower and intermediate frequency limits for the Fourier analysis, Figure 3—figure supplement 1) we sometimes obtained 3 significant classes, as indicated by the Calinski Harabasz criterion. When we obtain 3 classes, neurons in the intermediate class showed this combination of phasic and sustained features. Since we obtained two significant classes in most conditions (60% of tested parameter conditions), we conserved these two classes for subsequent analysis. However, we are now acknowledging the variety of activity patterns observed in this study in the last paragraph of the subsection “Classification of distinct neural signatures in the dorsal striatum during extended sequence training”.

We are now including further validation of our clustering approach by using a permutation test (presented in Figure 3—figure supplement 1) to demonstrate that the two classes provided by hierarchical clustering are indeed significantly different from each other.

3) Throughout the manuscript, the authors indicate that their analyses are "unbiased". It is unclear in what sense they are unbiased. The authors are making a number of choices toward particular analysis or criterion. Perhaps, use "objective"? or clarify in what sense unbiased.

We thank the reviewer for this suggestion; we are now using the word “objective” instead of “unbiased”.

4) In the correlation between neuronal activity and three phases of behavior (1st lever press latency, response rate, and port entry latency), the authors define the "execution" window as the window between the first and the last lever press. This is a variable window and therefore, will be highly anti-correlated with response rate even where there is no firing modulation. This analysis has to be done and reported carefully.

We understand the reviewer’s concern regarding this potential issue. Since the window of sequence execution (form 1^st^ to last lever press) is variable in duration, the number of spikes within this period was normalized by the sequence duration. This is now more clearly explained in the Materials and methods section: “Neural activity during execution of the lever press sequence was computed by normalizing the number of spikes between the first and last lever press to the sequence duration”.

5) It is unclear from the results "monitoring" is a good way to characterize the activity correlated with behaviors. It is possible that these activities are controlling behaviors. Why does the author use "monitoring"? That being said, a similar idea was proposed by Rueda-Orozco and Robbe, 2015. It would be good to cite this paper.

We initially used monitoring since our data do not clearly show how the neural activity we measured might control behavior. As mentioned above, it is not clear to what extent the activity we see represents the control or monitoring of behavior. Thus to take into account the comment from the reviewer, we have tried to mostly stay descriptive, stating that the activity in DMS and DLS remains correlated with performance after extended training. What these activity patterns could encode is discussed in the Discussion. And, we are now citing the Rueda-Orozco and Robbe paper which is indeed particularly relevant for the present study.

Reviewer #2:In this manuscript, Vandaele et al., recorded spiking activity simultaneously in the dorsomedial and dorsolateral regions of the striatum (DMS, DLS) during learning and performance of lever press sequence task. The authors wanted to examine the validity of a classical theory in the field: the DMS contributes to early phases of task learning while DLS contribute to performance once the task is habitual. The firing pattern of DMS and DLS population was distinct but did not evolve much during learning although they were more similar when recordings were performed after extensive training. The authors concluded that contrary to the classical view both regions are engaged throughout learning and long after learning.The general conclusion is interesting and could be novel enough to deserve publication in eLife. But in my opinion there many contradictory results and odds/unclear analyses. In particular, the authors developed a functional classification of the neurons (phasic, no-phasic; start, stop, middle) which is statistically contestable and is used extensively in the manuscript. More important (and again in my opinion) this analysis does not bring much more information or additional evidence in regard to the conclusion of the paper, as mentioned in the Abstract. I also find that some important observation which contradicts previous classical studies are not sufficiently emphasized. Finally, I have a strong technical concern I would like the author to clarify.

We appreciate the reviewer’s point of view. In point of fact, the reviewer is correct that the classification approach we used in the paper provides results that do not appreciably differ from the analysis of the mean activity of all task responsive neurons and from the decoding analysis. Each of the three approaches indicates that the activity in DMS and DLS is more different during 10 sessions of DT5 training than after 8 weeks of DT5 training. We were indeed surprised not to find relations between any of our approaches and the behavioral improvements over early training. We are viewing this agreement among three different approaches as a strength of our paper. Our preference would be to keep the classification approach since we suppose a common criticism should we remove this section would be that different typical response patterns (stop/start/sustained) proposed to reflect chunking/skill learning would have shown development across session 1 through 10, in possible contrast to the conclusions from average activity and from decoding. Our aim is to describe this classification approach as efficiently as possible so that it does not detract from the overall message of congruence found with these three approaches.

1) The authors claim that they use an unbiased classification of their neurons but in fact, many aspects of this classification are arbitrary and have unclear statistical ground. The authors decided to use the power in low (<1Hz) and "intermediate" (1-4Hz) frequency domains of each neuron spike train as features for hierarchical clustering. To the best of my knowledge, it is the first time such a method is used (at least the authors do not provide references). First, this choice of frequency is arbitrary. Second power in spike train is quite a tricky measure and is strongly sensitive to the total numbers of spikes recorded. I doubt this is a good method for low firing rates observed in the striatum. In support of my doubt, the average power spectrum shown in Figure 3—figure supplement 1 peak at the lowest frequency value (first bin after 0 Hz). I don't think it is statistically valid to use power spectrum analysis while a majority of neurons do not show clear oscillatory patterns. In other words, the clustering algorithm is applied to noisy data with arbitrary threshold. Third, it is not because a clustering algorithm returns an optimal number of clusters that it means that data can be statistically separated. This requires combining the clustering algorithm with a permutation of the data points (shuffling of the paired value) and bootstrapping. Without showing such statistical disambiguation the authors cannot reject the hypothesis that there is no discrete categories but rather a continuum of firing rate profiles. By looking at the data in Figure 3—figure supplement 1 it appears that there is significant overlap between the 2 main categories: the phasic neurons in the upper part of Figure 3—figure supplement 1D are more similar to non-phasic neurons in the upper panel of Figure 3—figure supplement 1E, than to phasic neurons in the lower part of Figure 3—figure supplement 1D.Without demonstrating that there are discrete functional categories of neurons (which is very unlikely by looking at the data in Figure 2), the authors should rewrite a large section of their manuscript and consider the possibility that striatum encodes continuously sensorimotor dynamics. This is an important point as this issue has been raised recently (a paper cited by the authors Sales-Carbonell et al., 2018 but see also Robbe, 2018).

We thank the reviewer for the suggestion of permutation test. We are now using this test to demonstrate that the two classes, phasic and non-phasic, provided by hierarchical clustering are indeed significantly different from each other (Figure 3—figure supplement 1F). We also agree with the reviewer that the choice of low (<1Hz) and intermediate (1-4Hz) ranges in frequency domain for the Fourier analysis is arbitrary, which we are now acknowledging in the Materials and methods subsection “Classification of distinct neural activity patterns”. Therefore, we manipulated these parameters to investigate the consistency in clustering results (Figure 3—figure supplement 1E). We varied the lower and higher frequency limits and assessed (1) the optimal number of cluster detected and (2) the distance between cluster means when 2 clusters were detected. We found that our results generalize well across ranges of frequency limit parameters (lower frequency limit: 0.3 to 1.0Hz; upper frequency limit: 2 to 8 Hz), the optimal number of clusters being 2 (60%) or 3 (33%) in 93% of parameter combinations. Furthermore, when 2 clusters were detected, cluster separability significantly departed from chance (permutation test) (Figure 3—figure supplement 1F). Finally, as can be seen in Author response image 3, Author response image 4 examples of hierarchical clustering providing 2 significant classes, the number and identity of Non Phasic and Phasic neurons was consistent across ranges of frequency limit parameters. In addition, the peaks in the frequency histogram depicted shows that around the same number of non phasic neurons were separated from the phasic neurons across all the parameter space, when two clusters were derived. Given that clustering results are not drastically impacted by the arbitrary choice of frequency range, we consider the range selected as reasonable. However, since this approach still requires arbitrary choices, we cannot characterize our analysis as “unbiased” and therefore replaced this word with “objective”.

**Author response image 3. respfig3:** 

To the best of our knowledge, this is the first time such method is used to separate phasic and non-phasic neurons. Thus, we are now better justifying our approach in the Materials and methods subsection “Classification of distinct neural activity patterns”. We also highlight that the Fourier analysis was not used to characterize inherent oscillatory activity, but, instead, as a tool to search for distinctions among the time courses of event-evoked spiking patterns. We agree that analysis of frequency spectrum is sensitive to difference in firing rate, which may impact hierarchical clustering. However, consistent clustering results across ranges of frequency domains suggest that the analysis provides robust results across a range of firing rates.

Also, instead of using oscillatory analysis as features characterizing neuronal activity, why not using the averaged and normalized peri-event histograms (data in Figure 2D, E) and compute PCA on these to characterize two or three values that capture most of the variability of the peri-event histograms (see Sales-Carbonnel, 2018).

We thank the reviewer for this suggestion. Hierarchical clustering using neurons’ coefficients in the first three principal components revealed 4 clusters of neurons as indicated by the Calinski Harabasz criterion. The heatmaps of individual neuron z-scores in these 4 clusters are shown in Author response image 4. Although some clusters were characterized by a specific pattern of activity (i.e. excitation at the end of the sequence in the first cluster (far left), excitations at the boundaries of the sequence in the last cluster (far right)) we observe significant overlap in activity patterns across clusters. This overlap is also visible by looking at the 3D scatter plot of neuron coefficients in PC1, PC2 and PC3, shown in Author response image 5. For this reason, we are not confident enough in the clustering results to use this approach in the present study. In contrast, the overlap we observe from using a Fourier analysis as a tool to separate “phasic” and “non-phasic” behavior-related activity patterns is low (please see Figure 3—figure supplement 1 and Figure 6—figure supplement 1).

The Fourier analysis was motivated by our initial observations that some neurons expressed sustained activity whereas others showed more transient modulation of activity along the behavioral sequence. We sought a way to distinguish these two groups that might be preferable to doing so “by eye”. The results of hierarchical clustering met this objective. We therefore feel more confident in the approach currently employed in this study.

**Author response image 4. respfig4:** 

**Author response image 5. respfig5:** 

2) In the Abstract, the authors stated that DMS and DLS are "engaged" during learning. This statement is a bit contradictory with the fact that firing patterns in both regions do not evolve while animals improve in the task. Again, this is especially contradictory as the authors emphasized the existence of star and stop neurons. Usually, a causal role in behavior is inferred when there is a change in neuronal activity that correlates with a change in task proficiency (e.g., Barnes et al., 2005, Jin and Costa, 2015).

We agree with the reviewer that changes in DMS and DLS activity patterns do not match with improvement in performance during early training. In fact, we acknowledge this in the Discussion: “[…]. Thus, we cannot conclude that the neural correlates we report here mediate the development of habit or the improvement in performance over time”. Yet, we observed in both DMS and DLS behavioral correlates with *performance* and our inactivation experiments reveal the involvement of these regions in DT5 performance. For instance, we agree with the reviewer comment below about a possible role of striatum in vigor and movement speed. To be more clear in our wording, we have tried to improve our emphasis on correlations of DMS and DLS activity with performance rather than skill or habitual learning.

3) The results of the inactivation experiments do not fit very well with the general spin of the paper as there is no strong effect on the number of lever press. The strongest effect seems to be an increase in the latency to 1st press and a decrease in response rate. Again, this is difficult to reconcile with striatal neurons starting or maintaining or stopping sequences. On the other hand, this result with work emphasizing a role of the striatum in vigor/movement speed (Kim et al., 2014, Rueda-Orozco et al., 2015, Panigrahi et al., 2015).

We agree with the reviewer that the results of the inactivation experiments are in line with prior work demonstrating the role of dorsal striatum in performance attributes, such as movement vigor. We are now citing the references suggested by the reviewer to emphasize this result in the ninth paragraph of the Discussion. In agreement with the reviewer, we do not think the striatal activity patterns we observe in our task are encoding the initiation and termination of the sequence. In fact, one paragraph in the Discussion is devoted to point out that “start” and “stop” activity in the dorsal striatum may largely represent cue-elicited activations, rather than motor sequence initiation-cessation signals. The supplementary figure supporting this conclusion is now presented as a main figure (Figure 4) to emphasize this important element of discussion, and we have tried to make this clearer in the revised Results (subsection “Classification of distinct neural signatures in the dorsal striatum during extended sequence training”, third paragraph).

4) One of the most surprising aspects of this data is that the population of DLS neurons fire continuously during the sequence. This is in plain contradiction with the start and stop hypothesis or the task bracketing hypothesis (especially it contradicts Martiros et al., 2018 paper that used a very similar lever press task). On the other hand, this result is very similar to Sales-Carbonnel et al., 2018 and Rueda-Orozco and Robbe). I am really surprised that the authors do not emphasize such an important result.

We thank the reviewer for suggesting this emphasis, as we have a similar view. We are now emphasizing this result, as follows: “Sustained excitation in DLS neurons is also consistent with studies recording DLS activity during locomotion tasks involving motor control (Rueda-Orozco and Robbe, 2015; Sales-carbonell et al., 2018).”

5) The title of the paper is "Distinct recruitment of dorsomedial and dorsolateral striatum erodes with extended training". And in the Abstract it is stated that DMS and DLS activity converged after extensive. First, the authors have no direct evidence for such convergence as they did not record from naive to extensive. During the first 10 sessions (at the end of which performance plateaued) there is no sign of convergence.

This statement in the Abstract is now corrected: “Although DMS and DLS were differentially involved during early training, their activity was similar following extended training”.

Second, it seems from Figure 2C that the increased similarity in coding arise from diminished modulation of both DLS and DMS population activity (Z score closer to zero). I am not sure that the authors can conclude in some kind of functional convergence while both regions are becoming less modulated.

The degree of modulation of DLS and DMS population activity is actually not decreased after extended training: the proportions of task-responsive neurons are similar. Z-scores are closer to zero in Figure 2C because the relative proportions of excitations and inhibitions are similar after extended training, but unbalanced across region during early training (Figure 2A). This can be observed by an analysis of normalized activity after separation of the population into the excited and inhibited TRN. This analysis reveals that DLS and DMS neurons are similarly modulated across early and extended training (Author response image 6).

**Author response image 6. respfig6:** 

6) The authors stated that they recorded in average 81 units per session in the DMS with 8 single electrodes implanted. That means they recorded routinely 10 well-isolated units per electrode. I doubt this is possible. Could the authors provide a view of these well-isolated clusters showing how they are separated from noise/MUA?

Ah, there has been a misunderstanding. The number of recorded units is reported across the entire group of rats (N=9 for the early training group). This point is now clearly specified in the first paragraph of the subsection “DMS and DLS neurons are differentially modulated in the DT5 procedure during early training but not after extended training”.

Author response image 7 illustrates the isolation of 2 different units using offline sorter.

**Author response image 7. respfig7:** 

Reviewer #3:The authors show that, although previous literature has shown the dorsomedial striatum (DMS) mediates goal-directed actions and the dorsolateral striatum (DLS) mediates habitual behaviors, the DMS remains engaged even after habit development. The recordings were done using a relatively novel behavioral task, which the researchers published on in 2017. The data are intriguing and represent an important challenge to current theories of habit formation, yet they are purely correlational. However, the lack of causal data might be balanced by a more thorough analysis of the recording data in correlation with specific behavioral events, and in this way the work could still be quite impactful as it challenges current theories of habit formation. We provide some suggestions for improvement below:• The authors present data from an early training and extended training group of rats, yet there is not a clear justification of the time points used in these groups and no discussion of how the reader should interpret these different timepoints in relation to our broader understanding of habit. If behavior stabilizes by the end of early training, yet neural activity changes by the time the extended training group is recorded, how are we to interpret these results? Is it possible the data recorded in the early training group are correlated with habit acquisition while the data recorded in the extended training group are due to later consolidation? A discussion on what the change in signal represents (presumably not changes in behavior) would benefit the overall interpretation of the observed changes in DMS and DLS signals.

In the Introduction we now seek to provide an improved justification of the time points used in this study with respect to habitual learning and skill consolidation. In the Discussion, we now bring up the notion of skill consolidation to explain alterations in sequence-related activity without concomitant change in behavior, and thank the reviewer for that suggestion. In the last paragraphs of the Discussion we provide some interpretations (although speculative) for the absence of changes during early training and the alteration in sequence-related activity after extended training (Discussion, last paragraph). We agree with the reviewer that interpretation is challenging given that no obvious neural correlates that could “explain” the improvements in performance across the first 10 sessions were identified, and we now seek to discuss more completely what the changes in signal might represent.

• The lack of data from a time point where the animals are not habitual limits the interpretations that can be made about signal changes that occur as habit forms.

We are now acknowledging this limitation in the Discussion, and we propose future experiences comparing dorsostriatal activity in the DT5 task with the free-running ratio tasks in which behavior remains under goal-directed control, to address this issue (Discussion, sixth paragraph).

• Figure 1C serves as the single piece of data to show animals are developing habit – however the authors note that the difference between valued and devalued responding is p=.06 which could be considered marginally significant. Looking at the individual data points reveals several animals that display noticeable decreases in lever pressing under the devalued condition, suggesting these animals are not yet habitual. Likewise, in the extended training group there are also multiple animals with noticeable decreases. The authors do make some attempt at analysis of the individual data in Supplementary Figure 12. We would suggest that this be moved into the main figures and analyzed further, as the results in the goal-directed vs. habitual rats are not identical. If the satiation probe is not reliably distinguishing differences, but is only good for distinguishing group averages (e.g. perhaps because the individual differences are simply due to an order effect of testing that needs to be counterbalanced in the group?), this limits the analyses that can be done, and should be noted.

As suggested by the reviewer, we moved Supplementary Figure 12 to a main figure (Figure 8) and discuss more deeply differences in activity in rats sensitive or insensitive to devaluation (subsection “Individual differences in sensitivity to outcome devaluation do not substantially correlate with dorsostriatal activity”). A major limitation in this analysis is that the numbers of neurons become too small for comparisons of proportions of distinct class types. However, we can compare the activity of task-responsive neurons irrespective of individual classes. Additional analysis are provided in the Figure 8—figure supplement 1. Although a testing order effect cannot be definitely excluded; however, a baseline reinforced training session was systematically conducted between two tests under extinction to minimize this caveat. In addition, we only observed less responding in the second test session compared to the first in half of the rats in both early training and extended training groups.

• Some rats are trained with sucrose and some with pellets. The authors previously showed that reward type mattered for habit formation (Vandaele et al., 2017). It is not clear in this manuscript which rats were trained using which reward and whether that affected any of the results.

An analysis comparing behavior and DLS/DMS activity in rats trained with liquid sucrose or grain-based pellet is now included and presented in the Figure 8—figure supplement 2 (see also subsection “Individual differences in sensitivity to outcome devaluation do not substantially correlate with dorsostriatal activity”, last paragraph). We observed that similar proportion of rats trained with liquid sucrose or grain based pellet developed habits in contradiction with prior findings. It is possible that neural recording and tethering prevented expression of habit in a subgroup of rats trained with sucrose, thereby explaining the trend reported in sensitivity to outcome devaluation, in line with our observation that tethered behavior, not surprisingly, does not exactly replicate untethered behavior for many rats. However, the low number of animals in each condition, combined with significant inter-individual variability in sensitivity to satiety induced devaluation, impedes robust conclusion about reward type effect in the present findings.

• Were recordings done during the probe sessions? These would be interesting to analyze and present, especially with respect to the individual differences in performance on the probe trials.

We wish there were more subjects in the two groups that were and were not sensitive to reward devaluation to be able to better address this question. In addition, although recording was done during probe sessions, the limited number of trials precludes reliable statistical analysis of the activity of individual neurons, and many rats responded for fewer than the maximum 10 trials Analyzing this limited number of trials did not provide compelling and meaningful results as shown in Author response image 8. While I would like to read into some of the activity representations, the neuron number is rather small and the SEM in many cases large.

**Author response image 8. respfig8:** 

• Although clearly somewhat rare, it would be interesting to see what happens in these recordings on trials in which the rats failed to complete the 5 presses (failure to complete ratio with 1 minute).

We agree with the reviewer. Unfortunately, omissions are too rare to allow any relevant analysis of the activity (at most 2 or 3 trials per sessions in a very limited number of sessions).

• We wonder whether other outcome measures might be better for evaluating habit. It would be informative to see response rates and CV within-sequence for the probe trials. It would also be good to show 1st lever press latency in Figure 1 for comparison with Figure 8.

Change in first lever press latency across early training and extended training sessions is now illustrated in Figure 1H. We agree that showing more measures could be useful for expressing the devaluation results. The additional measures suggested by the reviewer are depicted in Author response image 9. Normalized response rate and coefficient of variation of response rate were not significantly affected by devaluation in the early training group. We observed a significant effect of devaluation on normalized response rate on the extended training group, mainly due to the suppression of responding during the second trials in the devalued condition. Because of the discrete trial schedule, we feel that the usual measure used to assess habitual responding (normalized response rate) in free-running schedule does not generalize very well in this study since the discrete trials confine the ability to respond. Perhaps a useful measure is the number of trials upon which at least one lever press was recorded; this is now depicted in the manuscript in Figure 1I, and devalued responding was not significantly lower than valued. We also analyzed the mean first lever press latency during devaluation testing and found no significant differences between the valued and devalued conditions, despite a trend for the extended training group (Early training F_1,8_=1.65, p>0.2; Extended training F_1,7_=5.22, p=0.056).

**Author response image 9. respfig9:** 

• In Figure 8 the authors use 1st lever press latency as a measure of habit, however this measure is not presented as a reliable measure of habit previously in the manuscript. Also in Figure 8C-F – why are the same outcome measures not shown for both DMS and DLS?

The Figure 8 in the original manuscript (Figure 9 in the revised manuscript) now includes the same variables for both DMS and DLS inactivation. The decrease in first lever press latency across sessions is now presented in the first figure of the manuscript. Importantly, we try to present multiple behavioral measures, but seek not to claim they are a firm measure of habit per se, with the exception of the devaluation group means. We instead note the progressive changes in latencies and response rates to document performance improvements over the first 10 sessions. To address the reviewer’s point, we try to be consistent in this approach since we can’t say whether these measures are indicative of habitual responding.

• The characterization of neurons used throughout the manuscript lacks nuance in terms of what these neurons may be doing. For instance, in Figure 3 the authors categorize neurons as either phasic start or stop neurons depending on their peak activity during the task, however the stop neurons look like they may also be signaling the port entry – a more careful timing analysis of this data like an analysis of recordings during a port entry that doesn't follow sequenced nose poking would help explain the nature of this late increase.

We thank the reviewer for this suggestion of analysis; we are now more thoroughly describing the activity of start and Stop neurons (Figure 4) (subsection “Classification of distinct neural signatures in the dorsal striatum during extended sequence training”).In addition, we have expanded our discussion of what these start and stop neurons might signal (Discussion, fifth paragraph), since the timing of them suggests that they may not be motor sequence initiation and cessation signals as we initially hypothesized.

Additionally, several neurons within the heatmaps seem to fall under multiple categories (start and middle; middle and stop; start, middle, and stop) – however there is no mention of neurons that are active across multiple behavioral time points. In Figure 5, it is noticeable that similarly categorized DMS and DLS neurons do look very different from each other in terms of overall activity.

The reviewer is correct in that there are neurons that can be observed to show more than one profile. The classification approach places a given unit into only one group, and we think on average this approach is grouping the neurons into the class that best matches their maximal absolute activity changes. We agree that analysis of these hybrid neurons could be insightful; however in the case of this already complex data set, we feel that taking into account each combination of activity would give us pretty low n’s in some of these categories, as well as impede the interpretation of the data and the readability of the manuscript. To address the reviewer’s point we now discuss the overlap in activity patterns with some neurons combining several profile of activity (subsection “Classification of distinct neural signatures in the dorsal striatum during extended sequence training”, last paragraph).

• Interpretation of the results is limited by the fact that direct and indirect pathway neurons cannot be distinguished. Also, unless we are missing something, it does not appear that putative FSIs and unidentified neurons are included in the analyzed data.

The reviewer is right; we cannot distinguish MSNs from the direct and indirect pathways in the present study. This would require using a viral approach or D1-Cre and D2-Cre line rodents to combine neuronal recording with optotagging or a switch to calcium imaging. Our findings are comparable to a body of past work using electrophysiological recording in striatum in mouse, rat, and monkey that does not distinguish these pathways.

Putative FSIs and unidentified neurons were not included in the analysis. Since we used arbitrary thresholds to separate FSI from MSNs, we ensured exclusion of FSI by also removing neurons showing intermediate features (unidentified neurons). Although we recognize the important role of this subtype of neurons in habitual and automated behavior, we were not confident enough in the identification of putative FSI to include them in the analysis, and their small number in early training sessions prevent any reliable analysis (at most one neuron in DLS or DMS on a given session). The PSTH represents the average z-score of all putative-FSI across early training and extended training sessions in DLS and DMS. The reviewer can note on the heatmaps shown in Author response image 10, the sustained activity (excitation or inhibition) of putative-FSI during early training sessions and after extended training. This sustained activity clearly could be important in driving the sustained responses we report in the manuscript. However, we feel we cannot strongly make any claims about this small number of units without some type of genetic approach allowing us to reliably distinguish these units from MSNs.

**Author response image 10. respfig10:** 

Overall, this work has the potential to be very informative for understanding the role of striatal subregions in habitual responding. However, although the authors mention these findings are at odds with previous literature, they stop short of explaining why this may be or how we should view this circuit going forward in light of their findings, which ultimately limits the impact and significance of their work.

We thank the reviewer for this evaluation. To address this point, we are now putting more emphasis on differences and similarities with previous literature throughout the Discussion and provide in the last paragraph 2 possible explanations for our unexpected findings.